REGISTERED REPORT

# Registered Report: COT drives resistance to RAF inhibition through MAP kinase pathway reactivation

Vidhu Sharma[1], Lisa Young[1], Miguel Cavadas[2], Kate Owen[3], Reproducibility Project: Cancer Biology[1]*

[1]Applied Biological Materials, Richmond, Canada; [2]University of College Dublin, Dublin, Ireland; [3]University of Virginia, Charlottesville, United States

*For correspondence: tim@cos.io

Group author details:
Reproducibility Project: Cancer Biology*1* See page 31*

**Abstract** The Reproducibility Project: Cancer Biology seeks to address growing concerns about reproducibility in scientific research by conducting replications of selected experiments from a number of high-profile papers in the field of cancer biology. The papers, which were published between 2010 and 2012, were selected on the basis of citations and Altmetric scores (*Errington et al., 2014*). This Registered Report describes the proposed replication plan of key experiments from "COT drives resistance to RAF inhibition through MAPK pathway reactivation" by Johannessen and colleagues, published in *Nature* in 2010 (*Johannessen et al., 2010*). The key experiments to be replicated are those reported in Figures 3B, 3D-E, 3I, and 4E-F. In Figures 3B, D-E, RPMI-7951 and OUMS023 cells were reported to exhibit robust ERK/MEK activity concomitant with reduced growth sensitivity in the presence of the BRAF inhibitor PLX4720. MAP3K8 (COT/TPL2) directly regulated MEK/ERK phosphorylation, as the treatment of RPMI-7951 cells with a MAP3K8 kinase inhibitor resulted in a dose-dependent suppression of MEK/ERK activity (Figure 3I). In contrast, MAP3K8-deficient A375 cells remained sensitive to BRAF inhibition, exhibiting reduced growth and MEK/ERK activity during inhibitor treatment. To determine if RAF and MEK inhibitors together can overcome single-agent resistance, MAP3K8-expressing A375 cells treated with PLX4720 along with MEK inhibitors significantly inhibited both cell viability and ERK activation compared to treatment with PLX4720 alone, as reported in Figures 4E-F. The Reproducibility Project: Cancer Biology is collaboration between the Center for Open Science and Science Exchange and the results of the replications will be published in *eLife*.

## Introduction

Activation of the canonical mitogen activated protein kinase (MAPK) pathway occurs in response to the binding of growth factors, hormones, or neurotransmitters to receptor tyrosine kinase receptors located at the cell surface (*Dhomen and Marais, 2009*; *Lopez-Bergami et al., 2008*). In untransformed cells, receptor ligation induces the sequential activation of the small GTPase RAS, followed by RAF, MEK and ERK, which relays proliferative signals generated at the cell periphery into the nucleus to control cellular survival, differentiation and growth (*Inamdar et al., 2010*; *Panka et al., 2006*). Not surprisingly, dysregulation of MAPK signaling is common in many human cancers including melanoma. Mutations in the RAF and RAS genes (*Davies et al., 2002*; *Mercer and Pritchard, 2003*) that trigger constitutive activation of the MAPK pathway can result in uncontrolled cell proliferation, invasion, metastasis, survival and angiogenesis (*Panka et al., 2006*; *Sharma et al., 2006*; *Smalley et al., 2006*; *Smalley and Herlyn, 2006*).

BRAF is one of three members of the RAF family, which includes ARAF, BRAF, and CRAF (or RAF-1) (*Dhomen and Marais, 2009*). In melanoma, BRAF represents the most commonly mutated

gene in the MAPK signaling cascade where 90% of tumors carry a valine to glutamic acid transition at codon 600 (V600E) that renders BRAF constitutively active and hyperactivates the MAPK cascade (*Davies et al., 2002*; *Dhomen and Marais, 2009*; *Michaloglou et al., 2008*). While preclinical and clinical studies have shown that targeting BRAF (V600E) melanomas with the use of RAF-selective inhibitors results in initial tumor regression (*Fedorenko et al., 2015*; *Flaherty et al., 2010*; *Shtivelman et al., 2014*), responses to RAF inhibitors are transient, with acquired resistance triggering disease progression (*Shtivelman et al., 2014*). Although progress has been made in the development of drugs that target RAF, the clinical outcome regarding long-term usage and the mechanisms of acquired resistance warrants further evaluation. In their study, Johannessen and colleagues sought to identify kinases involved in mediating resistance to the RAF kinase inhibitor PLX4720 (*Johannessen et al., 2010*).

Using a kinase open reading frame (ORF) collection and a high throughput screening methodology, Johannessen and colleagues identified *MAP3K8* (the gene encoding cancer osaka thyroid (COT)/TPL2), as a driver of resistance to BRAF inhibition with PLX4720 (*Johannessen et al., 2010*). Johannessen and colleagues first examined basal MAP3K8 expression in multiple cell lines harboring the V600E mutation (*Johannessen et al., 2010*). As shown in Figure 3B and reported by others, RPMI-7951 and OUMS-23 cells were found to express high intrinsic levels of MAP3K8 compared to A375 cells where MAP3K8 was undetectable (*Johannessen et al., 2010*; *Paraiso et al., 2012*). RPMI-7951 and OUMS-23 cells also exhibited robust, undiminished ERK and MEK activity concomitant with reduced growth sensitivity in the presence of PLX4720 (Figure 3D–E; *Johannessen et al., 2010*). This is supported by additional findings demonstrating that RPMI-7981 cells treated with the closely related BRAF inhibitor PLX4032/vemurafenib (a successor of PLX4720) also remain refractory to inhibitor treatment as assessed by annexin V staining (*Paraiso et al., 2012*) and MTS assay (*Park et al., 2013*). However, others have reported RPMI-7981 cells as exhibiting modest sensitivity to PLX4720 (*Schayowitz et al., 2012*). In the latter case, ERK activity was reduced by 50% after incubation with inhibitor, although these differences in sensitivity may reflect the significantly shorter time course and experimental design used by Park and colleagues. Finally, in Figure 3I, Johannessen and colleagues determined that MAP3K8 kinase activity is required to regulate MEK/ERK activation in RPMI-7951 cells, findings that further confirm MAP3K8 is an essential upstream activator of the MEK-ERK signaling cascade (*George and Salmeron, 2009*; *Johannessen et al., 2010*). The key experiments outlined in Figures 3B,D,E, and 3I will be replicated in protocols 1, 2, 3, and 4.

Resistance to targeted agents, such as BRAF inhibitors, is a frequent cause of therapy failure, as noted above. Importantly, chronic BRAF inhibition can lead to cross-resistance to several BRAF-selective inhibitors, indicating that resistance is not likely to be overcome by switching to a new RAF inhibitor (*Corcoran et al., 2010*; *Villanueva et al., 2011*). It has been suggested previously that combination treatment with MEK and BRAF inhibitors may be useful in preventing the emergence of resistance or in overcoming resistance to single agent therapies targeting either molecule alone (*Corcoran et al., 2010*). To examine whether the combined use of RAF and MEK inhibitors bypass MAP3K8-driven resistance, Johannessen and colleagues ectopically expressed MAP3K8 in A375 melanoma cells before treatment with BRAF inhibitor (PLX4720) alone or in combination with the MEK inhibitors CI-1040 or AZD6244. As shown in Figures 4E and 4F, both viability and ERK activation was dramatically reduced in MAP3K8-expressing cells treated with either of the combination therapies, similar to cells ectopically expressing *MEK1*, which remained sensitive to PLX4720 (*Johannessen et al., 2010*). Similar results were obtained in RPMI-7951 cells expressing high basal levels of MAP3K8 treated with PLX4032 and a second MEK inhibitor AS703026 (*Park et al., 2013*). Interestingly, overexpression of constitutively active MEK (MEK1[DD]) resulted in increased sensitivity to BRAF inhibition combined with AZD6244, but not CI-1040 (*Johannessen et al., 2010*). These findings confirm that MAP3K8 is able to reactivate MAPK signaling despite BRAF inhibition and that targeting RAF and MEK in combination may be an effective anti-melanoma treatment strategy. These experiments will be replicated in Protocols 5 and 6.

# Materials and methods

## Protocol 1: MAPK pathway analysis in cells expressing elevated MAP3K8

This experiment assesses the effect the RAF inhibitor, PLX4720, has on the MAPK pathway, in cells expressing elevated MAP3K8, as analyzed via Western blot. It utilizes RPMI-7951 and OUMS-23 cells, which express a high level of MAP3K8 and A375 cells, which have undetectable levels. This protocol replicates the experiments reported in Figures 3B and 3E.

### Sampling

Experiment to be repeated a total of 4 times for a minimum power of 80%. The original data is qualitative, thus to determine an appropriate number of replicates to initially perform, sample sizes based on a range of potential variance was determined.

- See Power Calculations section for details.

Experiment has 3 cohorts:

- Cohort 1: A375 cells
- Cohort 2: RPMI-7951 cells
- Cohort 3: OUMS-23 cells

Each cohort has four conditions:

- Vehicle (DMSO)
- 10 μM PLX4720
- 1 μM PLX4720
- 0.1 μM PLX4720

Each condition will be probed with the following antibodies:

- pERK1/2 (T202/Y204)
- pERK1/2
- pMEK1/2 (S217/221)
- MEK1/2
- MAP3K8
- Actin

### Materials and reagents

| Reagent | Type | Manufacturer | Catalog # | Comments |
|---|---|---|---|---|
| RPMI medium with L-glutamine | Cell culture | Sigma | R8758 | Replaces Corning cat no. 10-040-CV. Communicated by authors. |
| MEM with L-glutamine | Cell culture | Sigma | M4655-500ML | Replaces Corning cat. No. 10-010-CV. Communicated by authors. |
| Fetal bovine serum (FBS) | Cell culture | Life Technologies | 12483-020 | Replaces Corning brand. Communicated by authors. |
| Pen/strep/glutamine | Cell culture | Abm | G255 | Replaces Corning brand. Communicated by authors. |
| A375 cells | Cell line | ATCC | CRL-1619 | Original brand not specified. |
| RPMI-7951 cells | Cell line | ATCC | HTB-66 | |
| OUMS-23 cells | Cell line | JCRB | JCRB1022 | Original brand not specified. |
| 6-well plates | Labware | Greiner bio-one | 657 160 | Original brand not specified. |
| Phosphate buffered saline (PBS) | Buffer | Sigma | D8537-500ML | Original brand not specified. |
| Trypsin | Cell culture | Sigma | T4049 | Original brand not specified. |

*Continued on next page*

*Continued*

| Reagent | Type | Manufacturer | Catalog # | Comments |
|---|---|---|---|---|
| 10 cm plates | Labware | CellStar | 664 160 | Original brand not specified. |
| PLX4720 | Inhibitor | Selleck Chemicals | S1152 | Replaces Symansis brand. |
| DMSO | Chemical | Sigma | D4540 | Original brand not specified. |
| NP-40 buffer | Buffer | Life tech | FNN0021 | Original brand not specified. |
| Protease inhibitors | Inhibitor | Roche | 04693116001 | Original catalog # not specified. |
| Phosphatase inhibitor cocktail I | Inhibitor | Sigma | P2850 | Replaces CalBioChem brand. |
| Phosphatase inhibitor cocktail II | Inhibitor | Sigma | P5726 | Replaces CalBioChem brand. |
| Cell scraper | Labware | Sarstedt | 83.1830 | Original brand not specified. |
| BCA kit | Reporter assay | Pierce | 23227 | Original catalog # not specified. Communicated by authors. |
| Dithiothreitol (DTT) | Chemical | Biobasic | DB0058 | Original brand not specified. |
| Sample buffer | Buffer | Abm | G031 | Replaces Invitrogen brand. |
| Protein molecular weight ladder | Western materials | Abm | G252, G494 | Original brand not specified. |
| 10% Tris/Glycine gel; 10 well, 1.0 mm thick | Western materials | Abm | Internal | Replaces Invitrogen brand. |
| Running buffer | Buffer | Abm | Internal | Original brand not specified. |
| Immobilon P | Western materials | Thermofisher | IPVH00010 | Original brand not specified. |
| Transfer buffer | Buffer | Abm | Internal | Original brand not specified. |
| Mouse anti-pERK1/2 (T202/Y204) (clone E10) antibody (clone E10) | Antibodies | Cell Signaling | 9106 | Use at 1:1000 dilution. Original catalog # not specified. |
| Rabbit anti-pMEK1/2 (S217/221) (clone 41G9) antibody | Antibodies | Cell Signaling | 9154 | Use at 1:11000 dilution. Original catalog # not specified. |
| Mouse anti-p44/42 MAPK (ERK1/2) (clone L34F12) antibody | Antibodies | Cell Signaling | 4696 | Use at 1:11000 dilution. Replaces catalog # 4695. Communicated by authors. |
| Rabbit anti-MEK1/2 (clone D1A5) antibody | Antibodies | Cell Signaling | 8727 | Use at 1:1000 dilution. Original catalog # not specified. |
| Rabbit anti-MAP3K8 (clone M-20) antibody | Antibodies | Santa Cruz | sc-720 | Use at 1:500 dilution. Communicated by authors. |
| Mouse anti-ß-Actin (clone C4) antibody | Antibodies | Santa Cruz | sc-47778 | Use at 1:100 – 1:1000 dilution. Original catalog # not specified. |
| Anti-rabbit IgG – HRP conjugated antibody | Antibodies | Cell Signaling | 7074 | Use at 1:1000 dilution. Original catalog # not specified. |
| Anti-mouse IgG – HRP conjugated antibody | Antibodies | Cell Signaling | 7076 | Use at 1:1000 dilution. Original catalog # not specified. |
| Chemiluminescent reagent | Western materials | Life Technologies | WP20005 | Replaces Pierce brand. |

## Procedure

Note:

- A375 cells maintained in RPMI medium supplemented with 10% FBS and 1% penicillin/strepto-mycin/L-glutamine at 37°C in a humidified atmosphere at 5% $CO_2$.
- RPMI-7951 and OUMS-23 cells maintained in MEM medium supplemented with 10% FBS and 1% penicillin/streptomycin/L-glutamine at 37°C in a humidified atmosphere at 5% $CO_2$.
- Cells will be sent for mycoplasma testing and STR profiling.

1. Plate 500,000 A375 cells, 750,000 RPMI-7951, and 750,000 OUMS-23 cells in 6-well plates and incubate for 24–36 hr to achieve log phase growth.
2. 24–36 hr after seeding treat cells with 0.1, 1, and 10 µM PLX4720 or DMSO. Incubate for 24 hr.
   a. Add drug directly to each well using a 1000X stock (in DMSO).
      i. Final DMSO concentration kept to 0.1%.
3. Wash cells with 1–2 ml ice-cold PBS and lyse in 1% NP-40 lysis buffer supplemented with 2X protease inhibitors and 1X phosphatase inhibitor cocktails I and II.
   a. Add ~100–200 µl 1% NP-40 lysis buffer to ensure that protein concentration is between 2–3 µg/µl.
   b. b. Scrape each plate with a rubber cell scraper, collect lysates, and clarify by centrifugation at max speed (table-top microfuge) at 4°C.
4. Determine protein concentration by BCA assay, normalize, reduce with DTT, and denature at 88°C.
5. Separate 35–50 $\mu$g of protein per lane on a 10% Tris/Glycine gel with protein ladder following replicating lab's standard protocol.
   a. Samples run per gel:
      i. Protein molecular weight marker
      ii. Vehicle (DMSO) treated A375 cells
      iii. 10 µM PLX4720 treated A375 cells
      iv. 1 µM PLX4720 treated A375 cells
      v. 0.1 µM PLX4720 treated A375 cells
      vi. Vehicle (DMSO) treated RPMI-7951 cells
      vii. 10 µM PLX4720 treated RPMI-7951 cells
      viii. 1 µM PLX4720 treated RPMI-7951 cells
      ix. 0.1 µM PLX4720 treated RPMI-7951 cells
      x. Vehicle (DMSO) treated OUMS-23 cells
      xi. 10 µM PLX4720 treated OUMS-23 cells
      xii. 1 µM PLX4720 treated OUMS-23 cells
      xiii. 0.1 µM PLX4720 treated OUMS-23 cells
6. Wet transfer with supplied wet-transfer cassette apparatus to immobilon P following replicating lab's standard protocol.
   a. Original transfer protocol was for 120min at 30–35 V at 4°C.
7. After transfer, block non-specific binding and immunoblot membrane with the following primary antibodies for 18 at 4°C following manufacturer recommendations:
   a. mouse anti-pERK1/2 (T202/Y204); use at 1:1000 dilution; 42, 44 kDa
   b. mouse anti-ERK1/2; use at 1:1000 dilution; 42, 44 kDa
   c. rabbit anti-pMEK1/2 (S217/221); use at 1:1000 dilution; 45 kDa
   d. rabbit anti-MEK1/2; use at 1:1000 dilution; 45 kDa
   e. rabbit anti- MAP3K8; use at 1:500 dilution; 52, 58 kDa
   f. mouse anti-ß-Actin; use at 1:100 - 1:1000 dilution; 43 kDa

**Protocol 1 Western Blot Antibody**

| Independent Gels | POI | | Loading Control | |
| --- | --- | --- | --- | --- |
| | Description | Working Conc. | Description | Working Conc. |
| 1 | Mouse anti-pERK1/2 (T202/Y204) (42, 44kDa) | 1:1000 | Rabbit anti-MEK1/2 (45 kDa) | 1:1000 |
| 2 | Rabbit anti-pMEK1/2 (S217/221) (45 kDa) | 1:1000 | Mouse anti-ERK1/2 (42, 44 kDa) | 1:1000 |
| 3 | Rabbit anti-MAP3K8 (52, 58 kDa) | 1:500 | Mouse anti-ß-Actin (43 kDa) | 1:100 – 1:1000 |

8. Apply appropriate HRP-linked secondary antibodies for 1 hr at RT with constant agitation, and then detect signal using chemiluminescence following manufacturer's instructions.
   a. Note: If a Li-COR Odyssey imaging system is available for use, IR Dye-labeled secondary antibodies and a low fluorescence membrane will be used instead, and images will be acquired following manufacturer's instructions.

9. Analyze bands with image analysis software and normalize to loading controls.
   a. pERK1/2 (T202/Y204) normalized to MEK1/2 (total).
   b. pMEK1/2 (S217/221) normalized to ERK1/2 (total).
   c. MAP3K8 normalized to Actin.
10. Repeat steps 1–9 independently three additional times.

## Deliverables:

- Data to be collected:
  - Full image western blot films of all immunoblots including ladder. (Compare to Figures 3B and 3E)
  - Raw data of band analysis and normalized bands for each sample.

## Confirmatory analysis plan

- Statistical Analysis of the Replication Data:
  - Two-way MANOVA of normalized pERK1/2 and pMEK1/2 levels of A375, RPMI-7951, and OUMS-23 cells with the following planned comparisons using the Bonferroni correction:
    - Planned contrast of normalized pERK1/2 levels from A375 cells treated with vehicle compared to cells treated with PLX4720 (all doses).
    - Planned contrast of normalized pERK1/2 levels from RPMI-7951 cells treated with vehicle compared to cells treated with PLX4720 (all doses).
    - Planned contrast of normalized pERK1/2 levels from OUMS-23 cells treated with vehicle compared to cells treated with PLX4720 (all doses).
    - Planned contrast of normalized pMEK1/2 levels from A375 cells treated with vehicle compared to cells treated with PLX4720 (all doses).
    - Planned contrast of normalized pMEK1/2 levels from RPMI-7951 cells treated with vehicle compared to cells treated with PLX4720 (all doses).
    - Planned contrast of normalized pMEK1/2 levels from OUMS-23 cells treated with vehicle compared to cells treated with PLX4720 (all doses).
- Meta-analysis of original and replication attempt effect sizes:
  - The replication data (mean and 95% confidence interval) will be plotted with the original reported data value plotted as a single point on the same plot for comparison.

## Known differences from the original study

The replication will not include the other BRAF (V600E) cell lines reported in the original paper. The original NP40 cell lysis buffer was composed of: 150 mM NaCl, 50 mM Tris pH 7.5, 2 mM EDTA pH 8, 25 mM NaF, and 1% NP-40. The replication will use a commercial formula, which has the following composition: 250 mM NaCl, 50 mM Tris pH 7.4, 5 mM EDTA, 50 mM NaF, 1 mM $Na_3VO_4$, and 1% NP-40. The western blots will use Actin, instead of Vinculin, which was reported in Figure 3B. All known differences are listed in the materials and reagents section above with the originally used item listed in the comments section. All differences have the same capabilities as the original and are not expected to alter the experimental design.

## Provisions for quality control

The cell line used in this experiment will undergo STR profiling to confirm its identity and will be sent for mycoplasma testing to ensure there is no contamination. All of the raw data, including the analysis files, will be uploaded to the project page on the OSF (https://osf.io/lmhjg/) and made publically available.

## Protocol 2: Determine the range of detection of the replicating lab's plate reader

This is a general protocol that determines the range of detection of the plate reader in order to calculate the required number of A375, RPMI-7951, and OUMS-23 cells to yield 90–95% confluency in 5 days for Protocols 3 and 5.

## Sampling

This experiment is performed a total of once with three cell lines (A375, RPMI-7951, and OUMS-23 cells).

Each cell line has 5 conditions to be performed with six technical replicates per experiment:

- A375 cells:
    - 1600 cells/well
    - 1400 cells/well
    - 1200 cells/well
    - 1000 cells/well
    - 800 cells/well
- RPMI-7951 cells
    - 3400 cells/well
    - 3200 cells/well
    - 3000 cells/well
    - 2800 cells/well
    - 2600 cells/well
- OUMS-23 cells
    - 3400 cells/well
    - 3200 cells/well
    - 3000 cells/well
    - 2800 cells/well
    - 2600 cells/well

## Materials and reagents

| Reagent | Type | Manufacturer | Catalog # | Comments |
|---|---|---|---|---|
| RPMI medium with L-glutamine | Cell culture | Sigma | R8758 | Replaces Corning cat no. 10-040-CV. |
| MEM with L-glutamine | Cell culture | Sigma | M4655-500ML | Replaces Corning cat. no. 10-010-CV. |
| FBS | Cell culture | Life Technologies | 12483-020 | Replaces Corning brand. |
| Pen/strep/glutamine | Cell culture | Abm | G255 | Replaces Corning brand. |
| A375 cells | Cell line | ATCC | CRL-1619 | Original brand not specified. |
| RPMI-7951 cells | Cell line | ATCC | HTB-66 | |
| OUMS-23 cells | Cell line | JCRB | JCRB1022 | Original brand not specified. |
| PBS | Buffer | Sigma | D8537-500ML | Original brand not specified. |
| Trypsin | Cell culture | Sigma | T4049 | Original brand not specified. |
| 10 cm plates | Labware | CellStar | 664 160 | Original brand not specified. |
| 96 well clear plates | Labware | Sarstedt | 83.3924 | Original brand not specified. |
| WST1 viability assay | Reporter assay | Roche | 11644807001 | Original catalog # not specified. |
| Microplate reader (420–480 nm) | Instrument | Molecular Devices | abm | Original brand not specified. |

## Procedure

Note:

- A375 cells maintained in RPMI medium supplemented with 10% FBS and 1% penicillin/streptomycin/L-glutamine at 37°C in a humidified atmosphere at 5% $CO_2$.
- RPMI-7951 and OUMS-23 cells maintained in MEM medium supplemented with 10% FBS and 1% penicillin/streptomycin/L-glutamine at 37°C in a humidified atmosphere at 5% $CO_2$.
- Cells will be sent for mycoplasma testing and STR profiling.

1. Plate 800 – 1600 A375 cells, 2600 – 3400 RPMI-7951 cells, and 2600 – 3400 OUMS-23 cells in 96 well plates with 100 µl of medium. Incubate for 5 days.
   a. Plate media alone (no cells) in columns 1 and 12.
   b. Plate cells in remaining wells (columns 2–11).
   c. Exclude plating in the first and last row to avoid edge effects and evaporation.
2. 5 days later estimate confluency and determine cell viability with the WST1 viability assay according to manufacturer's instructions. Briefly described:
   a. Add 11 µl/well reagent WST-1 (1:10 dilution).
   b. Incubate cells for 20–30 min.
   c. Shake thoroughly for 1 min on a shaker.
   d. Measure the absorbance against a background control as blank using a microplate reader at 420–480 nm. (If reference wavelength is to be determined, a filter >600 nm is recommended)
   e. Exclude rows A-H due to edge effects/evaporation, thus making each seeding six technical replicates.
   f. Calculate viability after background subtraction.
   g. Use starting cell numbers that give ~90–95% confluency in 5 days and is in the linear range of the viability assay.
      i. Original report used 1500 A375 cells, 3000 RPMI-7951 cells, and 3000 OUMS-23 cells.

## Deliverables

- Data to be collected:
  - Raw data and background subtracted absorbance at 420–480 nm.

## Confirmatory analysis plan

- n/a

## Known differences from the original study

All known differences are listed in the materials and reagents section above with the originally used item listed in the comments section. All differences have the same capabilities as the original and are not expected to alter the experimental design.

## Provisions for quality control

The cell line used in this experiment will undergo STR profiling to confirm its identity and will be sent for mycoplasma testing to ensure there is no contamination. This All of the raw data, including the analysis files, will be uploaded to the project page on the OSF (https://osf.io/lmhjg/) and made publically available.

## Protocol 3: PLX4720 growth inhibitory analysis in cells expressing elevated MAP3K8

This experiment assesses the effect the RAF inhibitor, PLX4720, has on cellular viability, in cells expressing elevated MAP3K8. It utilizes RPMI-7951 and OUMS-23 cells, which express a high level of MAP3K8 and A375 cells, which have undetectable levels. This protocol replicates the experiment reported in Figure 3D.

## Sampling

Experiment to be repeated a total of 3 times for a minimum power of 80%. The original data is from a single biological replicate, thus to determine an appropriate number of replicates to initially perform, sample sizes based on a range of potential variance was determined.

- See Power Calculations section for details.

Experiment has 3 cohorts:

- Cohort 1: A375 cells
- Cohort 2: RPMI-7951 cells
- Cohort 3: OUMS-23 cells

Each cohort has 9 conditions to be performed with six technical replicates per experiment:

- DMSO (vehicle)
- 100 µM PLX4720
- 10 µM PLX4720
- 1 µM PLX4720
- 0.1 µM PLX4720
- 0.01 µM PLX4720
- 0.001 µM PLX4720
- 0.0001 µM PLX4720
- 0.00001 µM PLX4720

## Materials and reagents

| Reagent | Type | Manufacturer | Catalog # | Comments |
|---|---|---|---|---|
| RPMI medium with L-glutamine | Cell culture | Sigma | R8758 | Replaces Corning cat no. 10-040-CV. |
| MEM with L-glutamine | Cell culture | Sigma | M4655-500ML | Replaces Corning cat. no. 10-010-CV. |
| FBS | Cell culture | Life Technologies | 12483-020 | Replaces Corning brand. |
| Pen/strep/glutamine | Cell culture | Abm | G255 | Replaces Corning brand. |
| A375 cells | Cell line | ATCC | CRL-1619 | Original brand not specified. |
| RPMI-7951 cells | Cell line | ATCC | HTB-66 | |
| OUMS-23 cells | Cell line | JCRB | JCRB1022 | Original brand not specified. |
| PBS | Buffer | Sigma | D8537-500ML | Original brand not specified. |
| Trypsin | Cell culture | Sigma | T4049 | Original brand not specified. |
| 10 cm plates | Labware | CellStar | 664 160 | Original brand not specified. |
| 96 well clear plates | Labware | Sarstedt | 83.3924 | Original brand not specified. |
| PLX4720 | Inhibitor | Selleck Chemicals | S1152 | Replaces Symansis brand. |
| DMSO | Chemical | Sigma | D4540 | Original brand not specified. |
| WST1 viability assay | Reporter assay | Roche | 11644807001 | Original catalog # not specified. |
| Microplate reader (420-480 nm) | Instrument | Molecular Devices | abm | Original brand not specified. |

## Procedure
Note:

- A375 cells maintained in RPMI medium supplemented with 10% FBS and 1% penicillin/streptomycin/L-glutamine at 37°C in a humidified atmosphere at 5% $CO_2$.
- RPMI-7951 and OUMS-23 cells maintained in MEM medium supplemented with 10% FBS and 1% penicillin/streptomycin/L-glutamine at 37°C in a humidified atmosphere at 5% $CO_2$.
- Cells will be sent for mycoplasma testing and STR profiling.

1. Plate number of A375, RPMI-7951, and OUMS-23 cells as determined in Protocol 2 in a 96 well plate with 90 µl of medium per well. Incubate for 24 hr.
   a. Plate media alone (no cells) in columns 1 and 12.
   b. Plate cells in remaining wells (columns 2–11).
   c. Exclude plating in the first and last row to avoid edge effects and evaporation.
   d. One plate is needed for each cell line.

2. Treat cells with 10 µl of 10X serial dilutions of PLX4720 to yield final dilutions of 100 µM to $10^{-5}$ µM (8 dilutions) (columns 3 through 10), or treat with DMSO (vehicle) control (columns 2 and 11). Incubate for 96 hr.
   a. Dilute stock of PLX4720 at 1000X final concentration of serial dilution stocks in DMSO (100 mM to 0.01 µM).
   b. Dilute 1000X serial dilution stocks 1:100 in complete growth medium to yield a 10X stock (1 mM to $10^{-4}$ µM) that is added directly to the 90 µl of cell/medium.
      i. Final DMSO concentration kept to 0.1%.
3. Determine cell viability with the WST1 viability assay according to manufacturer's instructions. Briefly described:
   a. Add 11 µl/well reagent WST-1 (1:10 dilution).
   b. Incubate cells for 20–30 min.
   c. Shake thoroughly for 1 min on a shaker.
   d. Measure the absorbance against a background control as blank using a microplate reader at 420–480 nm. (If reference wavelength is to be determined, a filter >600 nm is recommended)
   e. Exclude rows A-H due to edge effects/evaporation, thus making each cohort six technical replicates, except DMSO (vehicle), which has 12.
   f. Calculate viability as a percentage of control (DMSO (vehicle) cells) after background subtraction.
   g. Determine $GI_{50}$ value by fitting data using a nonlinear regression curve fit with a sigmoid dose-response curve (four-parameter log-logistic function).
4. Repeat steps 1–3 independently two additional times.

## Deliverables

- Data to be collected:
  - Raw data and background subtracted absorbance at 420–480 nm.
  - $GI_{50}$ values of each biological replicate.
  - Graph of average $GI_{50}$ values for each condition. (Compare to Figure 3D.)

## Confirmatory analysis plan

- Statistical Analysis of the Replication Data:
  - One way ANOVA of $GI_{50}$ values from A375, RPMI-7951, and OUMS-23 cells with the following planned comparisons using Fisher's LSD test.
- A375 cells compared to RPMI-7951 cells.
- A375 cells compared to OUMS-23 cells.
- Meta-analysis of original and replication attempt effect sizes:
  - The replication data (mean and 95% confidence interval) will be plotted with the original reported data value plotted as a single point on the same plot for comparison.

## Known differences from the original study

The replication will not include the other BRAF (V600E) cell lines reported in the original paper. All known differences are listed in the materials and reagents section above with the originally used item listed in the comments section. All differences have the same capabilities as the original and are not expected to alter the experimental design.

## Provisions for quality control

The cell line used in this experiment will undergo STR profiling to confirm its identity and will be sent for mycoplasma testing to ensure there is no contamination. The seeding density of each cell line was determined in Protocol 2. All of the raw data, including the analysis files, will be uploaded to the project page on the OSF (https://osf.io/lmhjg/) and made publically available.

## Protocol 4: MAPK pathway analysis after MAP3K8 inhibition in cells expressing elevated MAP3K8

This experiment assesses the effect a MAP3K8 kinase inhibitor has on the MAPK pathway, in cells expressing elevated MAP3K8, as analyzed via Western blot. It utilizes RPMI-7951 cells, which express a high level of MAP3K8. This protocol replicates the experiment reported in Figure 3I.

### Sampling

Experiment to be repeated a total of 8 times for a minimum power of 80%. The original data is qualitative, thus to determine an appropriate number of replicates to initially perform, sample sizes based on a range of potential variance was determined.

- See Power Calculations section for details.

Experiment has five conditions:

- Vehicle (DMSO) treated RPMI-7951 cells
- 20 µM MAP3K8 inhibitor treated RPMI-7951 cells
- 10 µM MAP3K8 inhibitor treated RPMI-7951 cells
- 5 µM MAP3K8 inhibitor treated RPMI-7951 cells
- 1 µM MAP3K8 inhibitor treated RPMI-7951 cells

Each condition will be probed with the following antibodies:

- pERK1/2 (T202/Y204)
- pERK1/2
- pMEK1/2 (S217/221)
- MEK1/2
- Vinculin

### Materials and reagents

| Reagent | Type | Manufacturer | Catalog # | Comments |
|---|---|---|---|---|
| MEM with L-glutamine | Cell culture | Sigma | M4655-500ML | Replaces Corningcat. no. 10-010-CV |
| FBS | Cell culture | Life Technologies | 12483-020 | Replaces Corning brand. |
| Pen/strep/glutamine | Cell culture | Abm | G255 | Replaces Corning brand. |
| RPMI-7951 cells | Cell line | ATCC | HTB-66 | |
| PBS | Buffer | Sigma | D8537-500ML | Original brand not specified. |
| Trypsin | Cell culture | Sigma | T4049 | Original brand not specified. |
| 10 cm plates | Labware | CellStar | 664 160 | Original brand not specified. |
| 6 well plates | Labware | Greiner bio-one | 657 160 | Original brand not specified. |
| MAP3K8 kinase inhibitor | Inhibitor | EMD | 616373 | |
| DMSO | Chemical | Sigma | D4540 | Original brand not specified. |
| NP-40 buffer | Buffer | Life Technologies | FNN0021 | Original brand not specified. |
| Protease inhibitors | Inhibitor | Roche | 04693116001 | Original catalog # not specified. |
| Phosphatase inhibitor cocktail I | Inhibitor | Sigma | P2850 | Replaces CalBioChem brand. |
| Phosphatase inhibitor cocktail II | Inhibitor | Sigma | P5726 | Replaces CalBioChem brand. |
| Cell scraper | Labware | Sasrstedt | 83.1830 | Original brand not specified. |

*Continued on next page*

*Continued*

| Reagent | Type | Manufacturer | Catalog # | Comments |
|---|---|---|---|---|
| BCA kit | Reporter assay | Pierce | 23227 | Original catalog # not specified. Communicated by authors. |
| DTT | Chemical | Biobasic | DB0058 | Original brand not specified. |
| Sample buffer | Buffer | abm | G031 | Replaces Invitrogen brand. |
| Protein molecular weight ladder | Western materials | abm | G252, G494 | Original brand not specified. |
| 10% Tris/Glycine gel; 10 well, 1.0 mm thick | Western materials | abm | internal | Replaces Invitrogen brand. |
| Running buffer | Buffer | abm | internal | Original brand not specified. |
| Immobilon P | Western materials | Thermofisher | IPVH00010 | Original brand not specified. |
| Transfer buffer | Buffer | abm | internal | Original brand not specified. |
| Mouse anti-pERK1/2 (T202/Y204) (clone E10) antibody | Antibodies | Cell Signaling | 9106 | Use at 1:1000 dilution. Original catalog # not specified. |
| Rabbit anti-pMEK1/2 (S217/221) (clone 41G9) antibody | Antibodies | Cell Signaling | 9154 | Use at 1:11000 dilution. Original catalog # not specified. |
| Mouse anti-p44/42 MAPK (ERK1/2) (clone L34F12) antibody | Antibodies | Cell Signaling | 4696 | Use at 1:11000 dilution. Replaces catalog # 4695. Communicated by authors. |
| Rabbit anti-MEK1/2 (clone D1A5) antibody | Antibodies | Cell Signaling | 8727 | Use at 1:1000 dilution. Original catalog # not specified. |
| Rabbit anti-Vinculin antibody | Antibodies | Sigma | V4139 | Use at 1:20,000 dilution. Original catalog # not specified. |
| Anti-rabbit IgG – HRP conjugated | Antibodies | Cell Signaling | 7074 | Use at 1:1000 dilution. Original catalog # not specified. |
| Anti-mouse IgG – HRP conjugated | Antibodies | Cell Signaling | 7076 | Use at 1:1000 dilution. Original catalog # not specified. |
| Chemiluminescent reagent | Western materials | Life Technologies | WP20005 | Replaces Pierce brand. |

## Procedure

Note:

- RPMI-7951 cells maintained in MEM medium supplemented with 10% FBS and 1% penicillin/streptomycin at 37°C in a humidified atmosphere at 5% $CO_2$.
- Cells will be sent for mycoplasma testing and STR profiling.

1. Plate 750,000 RPMI-7951 cells in 6-well plates and incubate for 24–36 hr to achieve log phase growth
2. Wash twice with 1X PBS and incubate overnight in serum-free growth medium.
3. Treat cells with 20, 10, 5, and 1 µM MAP3K8 inhibitor or DMSO for 1 hr.
   a. Add drug directly to each well.
   b. Make stocks of MAP3K8 inhibitor at 10 mM in DMSO. (500X dilution for 20 µM)
   c. Dilute stock of MAP3K8 to achieve 1000X final concentration of serial dilution stocks in DMSO (10 mM to 1000 µM).
   d. Final DMSO concentration kept to 0.2%.
4. Wash cells with 1–2 ml ice-cold PBS and lyse in 1% NP-40 lysis buffer (150 mM NaCl, 50 mM Tris pH 7.5, 2 mM EDTA pH 8, 25 mM NaF, and 1% NP-40) supplemented with 2X protease inhibitors and 1X phosphatase inhibitor cocktails I and II.
   a. Add ~100–200 µl 1% NP-40 lysis buffer to ensure that protein concentration is between 2–3 µg/µl.
   b. Scrape each plate with a rubber cell scraper, collect lysates, and clarify by centrifugation at max speed (table-top microfuge) at 4°C.
5. Determine protein concentration by BCA assay, normalize, reduce with DTT, and denature at 88°C.

6. Separate 35–50 $\mu$g of protein per lane on a 10% Tris/Glycine gel with protein ladder following replicating lab's standard protocol.
   a. Samples run per gel:
      i. Protein molecular weight marker
      ii. Vehicle (DMSO) treated RPMI-7951 cells
      iii. 20 µM MAP3K8 inhibitor treated RPMI-7951 cells
      iv. 10 µM MAP3K8 inhibitor treated RPMI-7951 cells
      v. 5 µM MAP3K8 inhibitor treated RPMI-7951 cells
      vi. 1 µM MAP3K8 inhibitor treated RPMI-7951 cells
7. Wet transfer with supplied wet-transfer cassette apparatus (120 min at 30–35 V at 4°C) to immobilon P following replicating lab's standard protocol.
8. After transfer, block non-specific binding and immunoblot membrane with the following primary antibodies for 18 hr at 4°C following manufacturer recommendations:
   a. mouse anti-pERK1/2 (T202/Y204); use at 1:1000 dilution; 42, 44 kDa
   b. mouse anti-ERK1/2; use at 1:1000 dilution; 42, 44 kDa
   c. rabbit anti-pMEK1/2 (S217/221); use at 1:1000 dilution; 45 kDa
   d. rabbit anti-MEK1/2; use at 1:1000 dilution; 45 kDa
   e. rabbit anti-Vinculin; use at 1:20,000 dilution; 116 kDa

**Protocol 4 Western blot antibody combinations**

| Independent Gels | POI | | Loading Control | |
| --- | --- | --- | --- | --- |
| | Description | Working Conc. | Description | Working Conc. |
| 1 | Mouse anti-pERK1/2 (T202/Y204) (42, 44 kDa) | 1:1000 | Rabbit anti-MEK1/2 (45 kDa) | 1:1000 |
| 2 | Rabbit anti-pMEK1/2 (S217/221) (45 kDa) | 1:1000 | Mouse anti-ERK1/2 (42, 44 kDa) | 1:1000 |
| 3 | | | Rabbit anti-Vinculin (116 kDa) | 1:20000 |

9. Apply appropriate HRP-linked secondary antibodies for 1 hr at RT with constant agitation, and then detect signal using chemiluminescence following manufacturer's instructions.
   a. Note: If a Li-COR Odyssey imaging system is available for use, IR Dye-labeled secondary antibodies and a low fluorescence membrane will be used instead, and images will be acquired following manufacturer's instructions.
10. Analyze bands with image analysis software, normalize to loading controls, and normalize each dose of MAP3K8 inhibitor to Vehicle (DMSO).
    a. pERK1/2 (T202/Y204) normalized to MEK1/2 (total).
    b. pMEK1/2 (S217/221) normalized to ERK1/2 (total).
11. Repeat steps 1–10 independently seven additional times.

## Deliverables

- Data to be collected:
  - Full image western blot films of all immunoblots including ladder. (Compare to Figure 3I)
  - Raw data of band analysis and normalized bands for each sample.

## Confirmatory analysis plan

- Statistical Analysis of the Replication Data:
  - One-way MANOVA of normalized pERK1/2 and pMEK1/2 levels of RPMI-7951 cells treated with MAP3K8 inhibitor with the following analysis using the Bonferroni correction:
    - One-way ANOVA of pERK1/2 levels of RPMI-7951 cells treated with MAP3K8 inhibitor.
      - One-sample $t$-test of pERK1/2 levels of 20 µM treated cells compared to 1 (vehicle treated cells).

- One-way ANOVA of pMEK1/2 levels of RPMI-7951 cells treated with MAP3K8 inhibitor.
  - One-sample $t$-test of pMEK1/2 levels of 20 μM treated cells compared to 1 (vehicle treated cells).
    - $IC_{50}$ values of normalized pERK1/2 and pMEK1/2 levels treated with vehicle or MAP3K8 inhibitor.
- Meta-analysis of original and replication attempt effect sizes:
  - The replication data (mean and 95% confidence interval) will be plotted with the original reported data value plotted as a single point on the same plot for comparison.

## Known differences from the original study

The original NP40 cell lysis buffer was composed of: 150 mM NaCl, 50 mM Tris pH 7.5, 2 mM EDTA pH 8, 25 mM NaF, and 1% NP-40. The replication will use a commercial formula, which has the following composition: 250 mM NaCl, 50 mM Tris pH 7.4, 5 mM EDTA, 50 mM NaF, 1 mM $Na_3VO_4$, and 1% NP-40. All known differences are listed in the materials and reagents section above with the originally used item listed in the comments section. All differences have the same capabilities as the original and are not expected to alter the experimental design.

## Provisions for quality control

The cell line used in this experiment will undergo STR profiling to confirm its identity and will be sent for mycoplasma testing to ensure there is no contamination. All of the raw data, including the analysis files, will be uploaded to the project page on the OSF (https://osf.io/lmhjg/) and made publically available.

## Protocol 5: Viability analysis following combinatorial MAPK pathway inhibition in cells expressing elevated MAP3K8

This experiment assesses the effect the RAF inhibitor, PLX4720, along with the MEK inhibitors, CI-1040 or AZD6244, has on cellular viability, in cells expressing MAP3K8. It utilizes A375 cells expressing MAP3K8, via ectopic expression of *MAP3K8*. This protocol replicates the experiment reported in Figure 4E.

### Sampling

Generation of A375 cells expressing MEK1, MEK1[DD], and MAP3K8 to be performed once.

Experiment (steps 4–6) to be repeated a total of 4 times for a minimum power of 80%. The original data is from a single biological replicate, thus to determine an appropriate number of replicates to initially perform, sample sizes based on a range of potential variance was determined.

- See Power Calculations section for details.

Experiment has 3 cohorts:

- Cohort 1: A375 cells expressing MEK1
- Cohort 2: A375 cells expressing MEK1[DD]
- Cohort 3: A375 cells expressing MAP3K8

Each cohort has 6 conditions to be done with six technical repeats per experiment:

- Untreated [additional control]
- DMSO (vehicle)
- 10 μM PLX4720
- 1 μM PLX4720
- 1 μM PLX4720 + 1 μM AZD6244
- 1 μM PLX4720 + 1 μM CI-1040

# Materials and reagents

| Reagent | Type | Manufacturer | Catalog # | Comments |
| --- | --- | --- | --- | --- |
| RPMI medium with L-glutamine | Cell culture | Sigma | R8758 | Replaces Corning cat no. 10-040-CV. |
| DMEM medium | Cell culture | Corning | 10-013 | Original brand not specified. |
| FBS | Cell culture | Life Technologies | 12483-020 | Replaces Corning brand. |
| Pen/strep/glutamine | Cell culture | Abm | G255 | Replaces Corning brand. |
| A375 cells | Cell line | ATCC | CRL-1619 | Original brand not specified. |
| 293T cells | Cell line | ATCC | CRL-11268 | |
| PBS | Buffer | Sigma | D8537-500ML | Original brand not specified. |
| Trypsin | Cell culture | Sigma | T4049 | Original brand not specified. |
| 10 cm plates | Labware | CellStar | 664 160 | Original brand not specified. |
| 6 cm plates | Labware | Biolite | 11825275 | Original brand not specified. |
| Nucleobond Maxiprep Kit | Kit | Macherey-Nagel | 740414 | Not originallyspecified |
| pLX-Blast-V5-MEK1 | DNA construct | Provided from original authors | | |
| pLX-Blast-V5-MEK1$^{DD}$ | DNA construct | Provided from original authors | | |
| pLX-Blast-V5-MAP3K8 | DNA construct | Provided from original authors | | |
| Δ8.9 (gag,pol) | DNA construct | Provided from original authors | | |
| VSV-G | DNA construct | Provided from original authors | | |
| FuGene HD transfection reagent | Transfection reagent | Promega | E2311 | Replaces FuGene6 Roche brand. |
| OptiMEM medium | Buffer | Life Tech | 51985034 | Original brand not specified. |
| 6-well plates | Labware | Greiner bio-one | 657 160 | Original brand not specified. |
| Polybrene | Cell culture | Sigma | H9268 | Original brand not specified. |
| Blasticidin | Cell culture | Invivogen | ant-bl-1 | Original brand not specified. |
| 96 well clear plates | Labware | Sarstedt | 83.3924 | Original brand not specified. |
| PLX4720 | Inhibitor | Selleck Chemicals | S1152 | Replaces Symansis brand. |
| AZD6244 | Inhibitor | Selleck Chemicals | S1008 | Original catalog # not specified. |
| CI-1040 | Inhibitor | Selleck Chemicals | S1020 | Replaces Shanghai Lechen International Trading Co. brand. |
| DMSO | Chemical | Sigma | D4540 | Original brand not specified. |
| WST1 viability assay | Reporter assay | Roche | 11644807001 | Original catalog # not specified. |
| Microplate reader (420–480 nm) | Instrument | Molecular Devices | abm | Original brand not specified. |

## Procedure

Note:

- A375 cells maintained in RPMI medium supplemented with 10% FBS and 1% penicillin/streptomycin/L-glutamine at 37˚C in a humidified atmosphere at 5% $CO_2$.
- 293T cells maintained in DMEM supplemented with 10% FBS and 1% penicillin/streptomycin at 37˚C in a humidified atmosphere at 5% $CO_2$.
- Cells will be sent for mycoplasma testing and STR profiling.

1. Grow and prepare endotoxin-free plasmid constructs according to the manufacturer's protocol for an endotoxin-free Plasmid Maxiprep Kit.
   a. Viral packaging vectors:
      i. Δ8.9 (gag,pol)

 ii. VSV-G
 b. DNA construct expression vectors:
 i. pLX-Blast-V5-MEK1
 ii. pLX-Blast-V5-MEK1$^{DD}$
 iii. pLX-Blast-V5-MAP3K8

2. Sequence expression plasmids to confirm identity and run on gel to confirm vector integrity. Use the following sequencing primers to confirm the identify of the pLX_CMV plasmids:
 a. pLX_CMV-ORF-fwd primer: 5'-CACCAAAATCAACGGGACTT-3'
 b. pLX-ORF-rev primer: 5'-AGGAGGAGAAAATGAAAGCC-3'

3. Produce MEK1, MEK1$^{DD}$, and MAP3K8 lentivirus:
 a. Seed 8 x 10$^5$ 293T cells per 6 cm dish, incubate.
 b. 24 hr later, transfect each plate of 293T cells by adding the following:
 i. 1 µg pLX-Blast-V5-MEK1, pLX-Blast-V5-MEK1$^{DD}$, or pLX-Blast-V5-MAP3K8
 ii. 900 ng Δ8.9 (gag,pol)
 iii. 100 ng VSV-G
 iv. 6 µl FuGene transfection reagent
 v. 94 µl OptiMem medium (free of FBS and Pen/Strep). Add OptiMem medium to the aliquots of DNA, then add the FuGene to the OptiMem/DNA mix.
 vi. Incubate 30 min at RT, then add to cells.
 c. Harvest virus 72 hr post-transfection, aliquot, and freeze at -80°C for at least 24 hr before using.
 i. Note: You will need to freeze down enough virus for both Protocols 5 and 6.

4. Titrate lentivirus:
 a. Seed 100,000 – 125,000 A375 cells per well in 6 well plates in 2 ml medium. Incubate for 24 hr in normal growth conditions.
 i. Seeding density is such that cells will be near confluent 3 days after removing virus.
 ii. Seed two wells per viral concentration for each virus (total wells = 30).
 b. Add polybrene (4–10 µg/ml final concentration) to plates, swirl to mix, then infect cells with varying concentrations of virus (1:5, 1:10, 1:12, 1:15, and 1:20) in duplicate (i.e. two wells per viral concentration).
 i. Thaw virus overnight at 4°C, or in a 37°C water bath, but without letting the virus get above 4°C.
 ii. Add virus to the wells and swirl to mix.
 iii. Spin 6-well plates at 2250 RPM for 30 min at 37°C.
 iv. Incubate cells overnight with virus, then change medium the following morning.
 c. Incubate for 24 hr, then remove medium and replace with growth medium with or without 10 µg/ml blasticidin.
 d. After 5–7 days of selection, count the cells in all wells and divide the counts with blasticidin by the non-selected control for each viral dilution.
 e. Use the lowest viral dilution that yields an 85–95% ratio of selected/non-selected cells (originally observed to be in the 1:8 – 1:15 range).

5. Infect A375 cells with viral supernatant:
 a. Seed 100,000 – 125,000 A375 cells per well in 6 well plates in 2 ml medium. Incubate for 24 hr in normal growth conditions.
 i. Seed three wells per virus. (total wells = 9).
 b. Add polybrene (4–10 µg/ml final concentration) to plates, swirl to mix, then infect cells with dilution determined by titration protocol (step 4 above).
 i. Thaw virus overnight at 4°C, or in a 37°C water bath, but without letting the virus get above 4°C.
 ii. Add virus to the wells and swirl to mix.
 1. Alternatively, make a 3X virus/polybrene mixture, such that the addition of 1 ml of 3X virus to 2 ml of cells/medium yields an appropriate dilution of virus and polybrene.
 iii. Spin 6-well plates at 2250 RPM for 30 min at 37°C.
 iv. Incubate cells overnight with virus, then change medium the following morning.
 c. Incubate for 24 hr, then remove medium and replace with growth medium with or without 10 µg/ml blasticidin.
 i. Two wells without selection and one well with blasticidin for each virus.
 d. After 5–7 days of selection, count cells in one well of no-blasticidin and the blasticidin-treated well to calculate infection efficiency.
 i. Divide the counts with blasticidin by the non-selected control.

6. After 48 hr trypsinize the experimental wells (one per virus), count, and plate number of A375 infected cells as determined in Protocol 2 in a 96 well plate in 90 µl. Incubate for 24 hr.
    a. Plate media alone (no cells) in columns 1,2,11, and 12.
    b. Plate cells in remaining wells (columns 3–10).
    c. Exclude plating in the first and last row to avoid edge effects and evaporation.
    d. One plate is needed for each of the A375 infected cells.
7. Treat cells with 10 µl of 10X dilutions of PLX4720 with or without AZD6244 or CI-1040 to yield appropriate final concentrations (columns 5 through 8), or treat with DMSO (vehicle) control (columns 4 and 9). Add 10 µl medium for untreated control (columns 3 and 10). Incubate for 96 hr.
    a. Dilute stock of PLX4720 at 1000X final concentration in DMSO (10 mM and 1 mM).
    b. Dilute stock of AZD6244 and CI-1040 at 1000X final concentration in DMSO (1 mM).
    c. Dilute 1000X dilution stocks 1:100 in complete growth medium to yield a 10X stock of appropriate treatment.
        i. Final DMSO concentration kept to 0.2%.
8. Determine cell viability with the WST1 viability assay according to manufacturer's instructions. Briefly described:
    a. Add 11 µl/well reagent WST-1 (1:10 dilution).
    b. Incubate cells for 20–30 min.
    c. Shake thoroughly for 1 min on a shaker.
        i. Measure the absorbance against a background control as blank using a microplate reader at 420–480 nm. (If reference wavelength is to be determined, a filter > 600 nm is recommended)
    d. Exclude rows A-H due to edge effects/evaporation, thus making each cohort six technical replicates, except DMSO (vehicle) and untreated control, which has 12.
    e. Calculate viability as a percentage of control (DMSO(vehicle) cells) after background subtraction.
9. Repeat steps 5–8 independently three additional times.

## Deliverables:

- Data to be collected:
    - Raw counts and titration percentages (step 2 above).
    - Raw counts and infection efficiency (step 3 above).
    - Raw data and background subtracted absorbance at 420–480 nm.
    - Relative viability of A375 cells expressing MEK1, MEK1[DD], and MAP3K8 as a percentage of DMSO treatment for each cell line. (Compare to Figure 4E.)
- Sample for additional protocol:
    - MEK1, MEK1[DD], and MAP3K8 lentivirus for further use (Protocol 6).

## Confirmatory analysis plan

- Statistical Analysis of the Replication Data:
    - One-way ANOVA of normalized viability of A375 cells expressing MAP3K8 treated with vehicle, 1 µM PLX4720, 1 µM PLX4720 + CI-1040, 1 µM PLX4720 + AZD6244, or 10 µM PLX4720 with the following planned comparisons using the Bonferroni correction:
        - 1 µM PLX4720 treatment compared to 1 µM PLX4720 + CI-1040.
        - 1 µM PLX4720 treatment compared to 1 µM PLX4720 + AZD6244.
        - 10 µM PLX4720 treatment compared to 1 µM PLX4720 + CI-1040.
        - 10 µM PLX4720 treatment compared to 1 µM PLX4720 + AZD6244.
    - Two-way ANOVA of normalized viability of A375 cells expressing MEK1 or MEK1[DD] treated with vehicle, 1 µM PLX4720, 1 µM PLX4720 + CI-1040, 1 µM PLX4720 + AZD6244, or 10 µM PLX4720.
- Meta-analysis of original and replication attempt effect sizes:
    - The replication data (mean and 95% confidence interval) will be plotted with the original reported data value plotted as a single point on the same plot for comparison.

## Known differences from the original study

All known differences are listed in the materials and reagents section above with the originally used item listed in the comments section. All differences have the same capabilities as the original and are not expected to alter the experimental design.

## Provisions for quality control

The cell line used in this experiment will undergo STR profiling to confirm its identity and will be sent for mycoplasma testing to ensure there is no contamination. The seeding density of the A375 cell line was determined in Protocol 2. Infection efficiency will be determined for each replicate. The expression of the kinase of interest will be assessed using antibodies against the V5 tag as well as MAP3K8 and MEK1 as described in Protocol 6. All of the raw data, including the analysis files, will be uploaded to the project page on the OSF (https://osf.io/lmhjg/) and made publically available.

### Protocol 6 MAPK pathway analysis following combinatorial MAPK pathway inhibition in cells expressing elevated MAP3K8

This experiment assesses the effect the RAF inhibitor, PLX4720, along with the MEK inhibitors, CI-1040 or AZD6244, has on the MAPK pathway, as analyzed via Western blot. It utilizes A375 cells expressing MAP3K8, via ectopic expression of *MAP3K8*. This protocol replicates the experiment reported in Figure 4F.

## Sampling

Experiment to be repeated a total of 3 times for a minimum power of 80%. The original data is qualitative, thus to determine an appropriate number of replicates to initially perform, sample sizes based on a range of potential variance was determined.

- See Power Calculations section for details.

Experiment has 3 cohorts:

- Cohort 1: A375 cells expressing MEK1
- Cohort 2: A375 cells expressing MEK1$^{DD}$
- Cohort 3: A375 cells expressing MAP3K8

Each cohort has 4 conditions:

- DMSO (vehicle)
- 1 µM PLX4720
- 1 µM PLX4720 + 1 µM AZD6244
- 1 µM PLX4720 + 1 µM CI-1040

Each condition will be probed with the following antibodies:

- pERK1/2 (T202/Y204)
- ERK1/2
- V5 [additional]
- Vinculin
- MEK1/2 [additional]
- MAP3K8 [additional]

## Materials and reagents

| Reagent | Type | Manufacturer | Catalog # | Comments |
|---|---|---|---|---|
| RPMI medium with L-glutamine | Cell culture | Sigma | R8758 | Replaces Corning cat no. 10-040-CV. |
| FBS | Cell culture | Life Technologies | 12483-020 | Replaces Corning brand. |

*Continued on next page*

*Continued*

| Reagent | Type | Manufacturer | Catalog # | Comments |
|---|---|---|---|---|
| Pen/strep/glutamine | Cell culture | Abm | G255 | Replaces Corning brand. |
| A375 cells | Cell line | ATCC | CRL-1619 | Original brand not specified. |
| PBS | Buffer | Sigma | D8537-500ML | Original brand not specified. |
| Trypsin | Cell culture | Sigma | T4049 | Original brand not specified. |
| 10 cm plates | Labware | CellStar | 664 160 | Original brand not specified. |
| 6 well plates | Labware | Greiner bio-one | 657 160 | Original brand not specified. |
| Polybrene | Cell culture | Sigma | H9268 | Original brand not specified. |
| Blasticidin | Cell culture | Invivogen | ant-bl-1 | Original brand not specified. |
| PLX4720 | Inhibitor | Selleck Chemicals | S1152 | Replaces Symansis brand. |
| AZD6244 | Inhibitor | Selleck Chemicals | S1008 | Original catalog # not specified. |
| CI-1040 | Inhibitor | Selleck Chemicals | S1020 | Replaces Shanghai Lechen International Trading Co. brand. |
| DMSO | Chemical | Sigma | D4540 | Original brand not specified. |
| NP-40 buffer | Buffer | Life technologies | FNN0021 | Original brand not specified. |
| Protease inhibitors | Inhibitor | Roche | 04693116001 | Original catalog # not specified. |
| Phosphatase inhibitor cocktail I | Inhibitor | Sigma | P2850 | Replaces CalBioChem brand. |
| Phosphatase inhibitor cocktail II | Inhibitor | Sigma | P5726 | Replaces CalBioChem brand. |
| Cell scraper | Labware | Sasrstedt | 83.1830 | Original brand not specified. |
| BCA kit | Reporter assay | Pierce | 23227 | Original catalog # not specified. Communicated by authors. |
| DTT | Chemical | Biobasic | DB0058 | Original brand not specified. |
| Sample buffer | Buffer | abm | G031 | Replaces Invitrogen brand. |
| Protein molecular weight ladder | Western materials | abm | G252, G494 | Original brand not specified. |
| 10% Tris/Glycine gel; 10 well, 1.0 mm thick | Western materials | abm | internal | Replaces Invitrogen brand. |
| Running buffer | Buffer | abm | internal | Original brand not specified. |
| Immobilon P | Western materials | Thermofisher | IPVH00010 | Original brand not specified. |
| Transfer buffer | Buffer | abm | internal | Original brand not specified. |
| Mouse anti-pERK1/2 (T202/Y204) (clone E10) antibody | Antibodies | Cell Signaling | 9106 | Use at 1:1000 dilution. Original catalog # not specified. |
| Mouse anti-p44/42 MAPK (ERK1/2) (clone L34F12) antibody | Antibodies | Cell Signaling | 4696 | Use at 1:11000 dilution. Replaces catalog # 4695. Communicated by authors. |
| Mouse anti-V5-HRP conjugated antibody | Antibodies | Invitrogen | R961-25 | Use at 1:5000 dilution. Original catalog # not specified. |
| Rabbit anti-Vinculin antibody | Antibodies | Sigma | V4139 | Use at 1:20,000 dilution. Original catalog # not specified. |
| Rabbit anti-MEK1/2 (clone D1A5) antibody | Antibodies | Cell Signaling | 8727 | Use at 1:1000 dilution. Original catalog # not specified. |
| Rabbit anti-MAP3K8 (clone M-20) antibody | Antibodies | Santa Cruz | sc-720 | Use at 1:500 dilution. Communicated by authors. |

*Continued on next page*

*Continued*

| Reagent | Type | Manufacturer | Catalog # | Comments |
|---|---|---|---|---|
| Anti-rabbit IgG – HRP conjugated antibody | Antibodies | Cell Signaling | 7074 | Use at 1:1000 dilution. Original catalog # not specified. |
| Anti-mouse IgG – HRP conjugated antibody | Antibodies | Cell Signaling | 7076 | Use at 1:1000 dilution. Original catalog # not specified. |
| Chemiluminescent reagent | Western materials | Life Technologies | WP20005 | Replaces Pierce brand. |

## Procedure

Note:

- A375 cells maintained in RPMI medium supplemented with 10% FBS and 1% penicillin/streptomycin at 37°C in a humidified atmosphere at 5% $CO_2$.
- Cells will be sent for mycoplasma testing and STR profiling.

1. Infect A375 cells with viral supernatant:
   a. Seed 100,000 – 125,000 A375 cells per well in 6 well plates in 2 ml medium. Incubate for 24 hr in normal growth conditions.
      i. Seed six wells per virus. (total wells = 18).
   b. Add polybrene (4–10 µg/ml final concentration) to plates, swirl to mix, then infect cells following Protocol 5, step 5 using dilution determined by titration protocol (Protocol 5, step 4).
   c. Incubate for 24 hr, then remove medium and replace with growth medium with or without 10 µg/ml blasticidin.
      i. Five wells without selection and one well with blasticidin for each virus.
   d. After 5–7 days of selection, count cells in one well of no-blasticidin and the blasticidin-treated well to calculate infection efficiency.
      i. Divide the counts with blasticidin by the non-selected control.
2. 72 hr after infection (96 hr after seeding) treat cells with DMSO or 1 µM PLX4720 with or without 1 µM AZD6244, 1 µM CI-1040, or DMSO. Incubate for 24 hr.
   a. Add drugs directly to each well using a 1000X stock (in DMSO).
      i. Final DMSO concentration kept to 0.2%.
3. Wash cells with 1–2 ml ice-cold PBS and lyse in 1% NP-40 lysis buffer (150 mM NaCl, 50 mM Tris pH 7.5, 2 mM EDTA pH 8, 25 mM NaF, and 1% NP-40) supplemented with 2X protease inhibitors and 1X phosphatase inhibitor cocktails I and II.
   a. Add ~100–200 µl 1% NP-40 lysis buffer to ensure that protein concentration is between 2–3 µg/µl.
   b. Scrape each plate with a rubber cell scraper, collect lysates, and clarify by centrifugation at max speed (table-top microfuge) at 4°C.
4. Determine protein concentration by BCA assay, normalize, reduce with DTT, and denature at 88°C.
5. Separate 35–50 $\mu$g of protein per lane on a 10% Tris/Glycine gel with protein ladder following replicating lab's standard protocol.
   a. Samples run per gel(s):
      i. Protein molecular weight marker
      ii. Vehicle (DMSO) treated A375 cells expressing MEK1
      iii. 1 µM PLX4720 treated A375 cells expressing MEK1
      iv. 1 µM PLX4720 and 1 µM AZD6244 treated A375 cells expressing MEK1
      v. 1 µM PLX4720 and 1 µM CI-1040 treated A375 cells expressing MEK1
      vi. Vehicle (DMSO) treated A375 cells expressing MEK1^DD
      vii. 1 µM PLX4720 treated A375 cells expressing MEK1^DD
      viii. 1 µM PLX4720 and 1 µM AZD6244 treated A375 cells expressing MEK1^DD
      ix. 1 µM PLX4720 and 1 µM CI-1040 treated A375 cells expressing MEK1^DD
      x. Vehicle (DMSO) treated A375 cells expressing MAP3K8
      xi. 1 µM PLX4720 treated A375 cells expressing MAP3K8
      xii. 1 µM PLX4720 and 1 µM AZD6244 treated A375 cells expressing MAP3K8
      xiii. 1 µM PLX4720 and 1 µM CI-1040 treated A375 cells expressing MAP3K8

6. Wet transfer with supplied wet-transfer cassette apparatus (120 min at 30–35 V at 4°C) to immobilon P following replicating lab's standard protocol.
7. After transfer, block non-specific binding and immunoblot membrane with the following primary antibodies for 18 hr at 4°C following manufacturer recommendations:
   a. mouse anti-pERK1/2 (T202/Y204); use at 1:1000 dilution; 42, 44 kDa
   b. mouse anti-ERK1/2; use at 1:1000 dilution; 42, 44 kDa
   c. mouse anti-V5-HRP; use at 1:5000 dilution; 45 kDa for MEK1, 52, 58 kDa for MAP3K8
   d. rabbit anti-Vinculin; use at 1:20,000 dilution; 116 kDa
   e. rabbit anti-MAP3K8; use at 1:500 dilution; 52, 58 kDa
   f. rabbit anti-MEK1/2; use at 1:1000 dilution; 45 kDa

**Protocol 6 Western blot antibody combinations**

| Independent Gels | POI | | Loading control | |
| --- | --- | --- | --- | --- |
| | Description | Working Conc. | Description | Working Conc. |
| 1 | Mouse anti-pERK1/2 (T202/Y204) (42, 44 kDa) | 1:1000 | Rabbit anti-MEK1/2 (45 kDa) | 1:1000 |
| 2 | Mouse anti-V5-HRP (45, 52, 58 kDa) | 1:5000 | Rabbit anti-Vinculin (116 kDa) | 1:20000 |
| 3 | Rabbit anti-MAP3K8 (52, 58 kDa) | 1:500 | Mouse anti-ERK1/2 (42, 44 kDa) | 1:1000 |

8. Apply appropriate HRP-linked secondary antibodies for 1 hr at RT with constant agitation, and then detect signal using chemiluminescence following manufacturer's instructions.
   a. Note: If a Li-COR Odyssey imaging system is available for use, IR Dye-labeled secondary antibodies and a low fluorescence membrane will be used instead, and images will be acquired following manufacturer's instructions.
9. Analyze bands with image analysis software and normalize to loading controls.
   a. pERK1/2 (T202/Y204) normalized to MEK1/2 (total).
   b. V5 (tag on exogenous proteins) normalized to Vinculin.
10. Repeat steps 1–9 independently two additional times.

## Deliverables:

- Data to be collected:
  - Raw counts and infection efficiency (step 1 above).
  - Full image western blot films of all immunoblots including ladder. (Compare to Figures 2A and 4F.)
  - Raw data of band analysis and normalized bands for each sample.

## Confirmatory analysis plan

- Statistical Analysis of the Replication Data:
  - One-way ANOVA of normalized pERK1/2 levels of A375 cells expressing MAP3K8 treated with vehicle, PLX4720, PLX4720 + CI-1040, or PLX4720 + AZD6244 with the following planned comparisons using the Bonferroni correction:
    - PLX4720 treatment compared to PLX4720 + CI-1040.
    - PLX4720 treatment compared to PLX4720 + AZD6244.
  - Two-way ANOVA of normalized pERK1/2 levels of A375 cells expressing MEK1 or MEK1$^{DD}$ treated with vehicle, PLX4720, PLX4720 + CI-1040, or PLX4720 + AZD6244.
- Meta-analysis of original and replication attempt effect sizes:
  - The replication data (mean and 95% confidence interval) will be plotted with the original reported data value plotted as a single point on the same plot for comparison.

## Known differences from the original study

The original NP40 cell lysis buffer was composed of: 150 mM NaCl, 50 mM Tris pH 7.5, 2 mM EDTA pH 8, 25 mM NaF, and 1% NP-40. The replication will use a commercial formula, which has the following composition: 250 mM NaCl, 50 mM Tris pH 7.4, 5 mM EDTA, 50 mM NaF, 1 mM $Na_3VO_4$, and 1% NP-40. All known differences are listed in the materials and reagents section above with the originally used item listed in the comments section. All differences have the same capabilities as the original and are not expected to alter the experimental design.

## Provisions for quality control

The cell line used in this experiment will undergo STR profiling to confirm its identity and will be sent for mycoplasma testing to ensure there is no contamination. Infection efficiency will be determined for each replicate. The expression of the kinase of interest will be assessed using antibodies against the V5 tag as well as MAP3K8 and MEK1. All of the raw data, including the analysis files, will be uploaded to the project page on the OSF (https://osf.io/lmhjg/) and made publically available.

## Power Calculations

For additional details on power calculations, please see analysis scripts and associated files on the Open Science Framework:

   https://osf.io/sptzv/

## Protocol 1

The original data presented is qualitative (images of Western blots). We used Image Studio Lite v. 4.0.21 (LI-COR) to perform densitometric analysis of the presented bands to quantify the original effect size where possible. To identify a suitable sample size, power calculations were performed using different levels of relative variance.

Summary of estimated original data reported in Figure 3E:

| Cell line | PLX4720 concentration (μM) | Normalized pMEK/ERK | Normalized pERK/MEK |
|---|---|---|---|
| A375 | 0 (DMSO) | 1.000 | 1.000 |
| | 10 | 0.00212 | 0.0147 |
| | 1 | 0.00283 | 0.0254 |
| | 0.1 | 0.0981 | 0.00283 |
| RPMI-7951 | 0 (DMSO) | 1.000 | 1.000 |
| | 10 | 0.827 | 1.849 |
| | 1 | 0.380 | 1.460 |
| | 0.1 | 0.471 | 1.022 |
| OUMS-23 | 0 (DMSO) | 1.000 | 1.000 |
| | 10 | 0.0579 | 0.926 |
| | 1 | 0.197 | 1.042 |
| | 0.1 | 1.100 | 1.068 |

## Test family

- *F* test, ANOVA: Fixed effects, special, main effects and interactions, Bonferorni's correction: alpha error = 0.00833

Power Calculations performed with G*Power software, version 3.1.7 (*Faul et al., 2007*).
   Partial $\eta^2$ calculated from (*Lakens, 2013*).
   Comparisons are between DMSO and all PLX4720 doses (10, 1, and 0.1 μM)

| Antibody | Cell line | F test statistic | Partial $\eta^2$ | Effect size $f$ | A priori power | Total sample size |
|---|---|---|---|---|---|---|
| **2% variance** | | | | | | |
| pMEK | A375 | F(1,8) = 20781 | 0.99962 | 50.9670 | 99.9% | 12 (4 groups) |
| | RPMI-7951 | F(1,8) = 2128.4 | 0.99626 | 16.3112[1] | 80.0% | 12 (4 groups) |
| | OUMS-23 | F(1,8) = 3001.0 | 0.99734 | 19.3683[1] | 80.0% | 12 (4 groups) |
| pERK | A375 | F(1,8) = 21841 | 0.99963 | 52.2504 | 99.9% | 12 (4 groups) |
| | RPMI-7951 | F(1,8) = 582.85 | 0.98646 | 8.53562[1] | 80.0% | 12 (4 groups) |
| | OUMS-23 | F(1,8) = 0.8080 | 0.09173 | 0.31780[1] | 80.0% | 12 (4 groups) |
| **15% variance** | | | | | | |
| pMEK | A375 | F(1,8) = 369.44 | 0.97880 | 6.79560 | 99.9% | 12 (4 groups) |
| | RPMI-7951 | F(1,8) = 37.839 | 0.82547 | 2.17482[1] | 80.0% | 12 (4 groups) |
| | OUMS-23 | F(1,8) = 53.352 | 0.86960 | 2.58244[1] | 80.0% | 12 (4 groups) |
| pERK | A375 | F(1,8) = 388.28 | 0.97981 | 6.96672 | 99.9% | 12 (4 groups) |
| | RPMI-7951 | F(1,8) = 10.362 | 0.56431 | 1.13808[1] | 80.0% | 12 (4 groups) |
| | OUMS-23 | F(1,8) = 0.01436 | 0.001792 | 0.04237[1] | 80.0% | 12 (4 groups) |
| **28% variance** | | | | | | |
| pMEK | A375 | F(1,8) = 106.68 | 0.92984 | 3.64050 | 99.9% | 12 (4 groups) |
| | RPMI-7951 | F(1,8) = 10.859 | 0.57581 | 1.16509[1] | 80.0% | 12 (4 groups) |
| | OUMS-23 | F(1,8) = 15.312 | 0.65682 | 1.38345[1] | 80.0% | 12 (4 groups) |
| pERK | A375 | F(1,8) = 111.43 | 0.93302 | 3.73217 | 99.9% | 12 (4 groups) |
| | RPMI-7951 | F(1,8) = 2.9737 | 0.27099 | 0.60969[1] | 80.0% | 12 (4 groups) |
| | OUMS-23 | F(1,8) = 0.04122 | 0.000515 | 0.02270[1] | 80.0% | 12 (4 groups) |
| **40% variance** | | | | | | |
| pMEK | A375 | F(1,8) = 51.952 | 0.86656 | 2.54835 | 99.9% | 12 (4 groups) |
| | RPMI-7951 | F(1,8) = 5.3211 | 0.39945 | 0.81556[1] | 80.0% | 12 (4 groups) |
| | OUMS-23 | F(1,8) = 7.5026 | 0.48396 | 0.96842[1] | 80.0% | 12 (4 groups) |
| pERK | A375 | F(1,8) = 54.602 | 0.87221 | 2.61252 | 99.9% | 12 (4 groups) |
| | RPMI-7951 | F(1,8) = 1.4571 | 0.15408 | 0.42678[1] | 80.0% | 12 (4 groups) |
| | OUMS-23 | F(1,8) = 0.00202 | 0.000252 | 0.01589[1] | 80.0% | 12 (4 groups) |

[1] This is the calculated effect size using the originally reported value with the indicated variance. Unlike the power calculations to determine sample size, the aim of these sensitivity calculations are not to detect the original effect

size, but to understand what effect size could be detected with 80% power and the indicated total sample size of 12. The detectable effect size is 1.29189.

In order to produce quantitative replication data, we will run the experiment three times. Each time we will quantify band intensity. We will determine the standard deviation of band intensity across the biological replicates and combine this with the reported value from the original study to simulate the original effect size. We will use this simulated effect size to determine the number of replicates necessary to reach a power of at least 80%. We will then perform additional replicates, if required, to ensure that the experiment has more than 80% power to detect the original effect.

## Protocol 2
Not applicable

## Protocol 3
The original data is from a single biological replicate. To identify a suitable sample size, power calculations were performed using different levels of relative variance.

Summary of estimated original data reported in Figure 3D:

| Cell line | GI$_{50}$ ($\mu$M) |
| --- | --- |
| A375 | 0.2276 |
| RPMI-7951 | 10.862 |
| OUMS-23 | 8.731 |

Test family

- 2 tailed $t$ test, difference between two independent means, Fisher's LSD: alpha error = 0.05

Power Calculations performed with G*Power software, version 3.1.7 (*Faul et al., 2007*).

| Group 1 | Group 2 | Effect size $d$ | A priori power | Group 1 sample size | Group 2 sample size |
| --- | --- | --- | --- | --- | --- |
| 2% variance | | | | | |
| A375 | RPMI-7951 | 69.2139 | 99.9% | 2 | 2 |
| A375 | OUMS-23 | 55.3442 | 99.9% | 2 | 2 |
| 15% variance | | | | | |
| A375 | RPMI-7951 | 68.3898 | 99.9% | 2 | 2 |
| A375 | OUMS-23 | 7.37923 | 93.3% | 2 | 2 |
| 28% variance | | | | | |
| A375 | RPMI-7951 | 4.94385 | 99.2% | 3 | 3 |
| A375 | OUMS-23 | 3.95316 | 94.4% | 3 | 3 |
| 40% variance | | | | | |
| A375 | RPMI-7951 | 3.46070 | 88.0% | 3 | 3 |
| A375 | OUMS-23 | 2.76721 | 90.0% | 4 | 4 |

In order to produce quantitative replication data, we will run the experiment three times. Each time we will calculate the GI$_{50}$ value. We will determine the standard deviation of GI$_{50}$ values across the biological replicates and combine this with the reported value from the original study to simulate the original effect size. We will use this simulated effect size to determine the number of replicates necessary to reach a power of at least 80%. We will then perform additional replicates, if required, to ensure that the experiment has more than 80% power to detect the original effect.

## Protocol 4

The original data does not indicate the error associated with multiple biological replicates. To identify a suitable sample size, power calculations were performed using different levels of relative variance.

## Summary of original data reported in Figure 3I:

| Cell line | MAP3K8 inhibitor concentration (μM) | Normalized pMEK/ERK | Normalized pERK/MEK |
|---|---|---|---|
| RPMI-7951 | 0 (DMSO) | 1.00 | 1.00 |
| | 1 | 1.35 | 1.00 |
| | 5 | 1.15 | 0.92 |
| | 10 | 0.70 | 0.75 |
| | 20 | 0.36 | 0.71 |

$IC_{50}$ values performed with R software, version 3.2.1 (*Team, 2015*).

| $IC_{50}$ value of normalized pMEK values | $IC_{50}$ value of normalized pERK values |
|---|---|
| 9.4725 | 6.4091 |

## Test family

- ▪ *F* test, ANOVA: Fixed effects, omnibus, one-way, Bonferroni's correction: alpha error = 0.025

Power Calculations performed with G*Power software, version 3.1.7 (*Faul et al., 2007*).
ANOVA F test statistic and partial $\eta^2$ performed with R software, version 3.2.1 (*Team, 2015*).
Partial $\eta^2$ calculated from (*Lakens, 2013*).

| Groups | Antibody | F test statistic | Partial $\eta^2$ | Effect size $f$ | A priori power | Total sample size |
|---|---|---|---|---|---|---|
| 2% variance | | | | | | |
| MAP3K8 inhibitor (1, 5, 10, 20 μM) | pMEK | $F(3,8) = 1583.7$ | 0.99832 | 24.3697 | 99.9% | 8 (4 groups) |
| | pERK | $F(3,8) = 195.33$ | 0.98653 | 8.55863 | 99.9% | 8 (4 groups) |
| 15% variance | | | | | | |
| MAP3K8 inhibitor (1, 5, 10, 20 μM) | pMEK | $F(3,8) = 28.155$ | 0.91348 | 3.24931 | 96.8% | 8 (4 groups) |
| | pERK | $F(3,8) = 3.4726$ | 0.56563 | 1.14114 | 83.6% | 16 (4 groups) |
| 28% variance | | | | | | |
| MAP3K8 inhibitor (1, 5, 10, 20 μM) | pMEK | $F(3,8) = 8.0801$ | 0.75186 | 1.74070 | 94.7% | 12 (4 groups) |
| | pERK | $F(3,8) = 0.9966$ | 0.27205 | 0.61133 | 80.8% | 40 (4 groups) |
| 40% variance | | | | | | |
| MAP3K8 inhibitor (1, 5, 10, 20 μM) | pMEK | $F(3,8) = 3.9593$ | 0.59754 | 1.21849 | 88.7% | 16 (4 groups) |
| | pERK | $F(3,8) = 0.4883$ | 0.15478 | 0.42793 | 80.5% | 76 (4 groups) |

## Test family

- *t*-test: Means: Difference from constant (one sample case): Bonferroni's correction: alpha error = 0.025

Power Calculations performed with G*Power software, version 3.1.7 (*Faul et al., 2007*).

| MAP3K8 inhibitor dose | Constant (vehicle) | Antibody | Effect size *d* | A priori power | Sample size per group |
|---|---|---|---|---|---|
| 2% variance | | | | | |
| 20 μM | 0 μM | pMEK | 88.8889 | 99.9% | 3 |
| | | pERK | 20.4225 | 99.9% | 3 |
| 15% variance | | | | | |
| 20 μM | 0 μM | pMEK | 11.8519 | 99.9% | 3 |
| | | pERK | 2.72300 | 80.2% | 4 |
| 28% variance | | | | | |
| 20 μM | 0 μM | pMEK | 6.34921 | 95.1% | 3 |
| | | pERK | 1.45875 | 86.6% | 8 |
| 40% variance | | | | | |
| 20 μM | 0 μM | pMEK | 4.44444 | 99.2% | 4 |
| | | pERK | 1.02113 | 81.1% | 12 |

In order to produce quantitative replication data, we will run the experiment eight times. Each time we will quantify band intensity. We will determine the standard deviation of band intensity across the biological replicates and combine this with the reported value from the original study to simulate the original effect size. We will use this simulated effect size to determine the number of replicates necessary to reach a power of at least 80%. We will then perform additional replicates, if required, to ensure that the experiment has more than 80% power to detect the original effect.

## Protocol 5

The original data is from a single biological replicate. To identify a suitable sample size, power calculations were performed using different levels of relative variance.

## Summary of original data reported in Figure 4E (provided by authors)

| Cell line | Drug(s) | Relative viability | Technical replicate stdev |
|---|---|---|---|
| MEK1 | DMSO | 100 | 5.09 |
| | PLX4720 (10 μM) | 5.78 | 0.67 |
| | PLX4720 (1 μM) | 11.32 | 0.97 |
| | PLX4720 (1 μM) + CI-1040 (1 μM) | 10.13 | 0.60 |
| | PLX4720 (1 μM) + AZD6244 (1 μM) | 10.92 | 0.91 |
| MEK1DD | DMSO | 96.96 | 9.40 |
| | PLX4720 (10 μM) | 97.56 | 6.25 |
| | PLX4720 (1 μM) | 92.10 | 8.23 |
| | PLX4720 (1 μM) + CI-1040 (1 μM) | 102.24 | 8.55 |
| | PLX4720 (1 μM) + AZD6244 (1 μM) | 42.44 | 2.74 |

*Continued on next page*

*Continued*

| Cell line | Drug(s) | Relative viability | Technical replicate stdev |
|---|---|---|---|
| MAP3K8 | DMSO | 100 | 10.66 |
| | PLX4720 (10 µM) | 43.44 | 3.48 |
| | PLX4720 (1 µM) | 62.81 | 5.05 |
| | PLX4720 (1 µM) + CI-1040 (1 µM) | 13.42 | 2.42 |
| | PLX4720 (1 µM) + AZD6244 (1 µM) | 12.52 | 1.65 |

## Test family

- *F* test, ANOVA: Fixed effects, omnibus, one-way, alpha error = 0.05.

Power Calculations performed with G*Power software, version 3.1.7 (*Faul et al., 2007*).
ANOVA F test statistic and partial $\eta^2$ performed with R software, version 3.2.1 (*Team, 2015*).

## MAP3K8 values

| Groups | F test statistic | Partial $\eta^2$ | Effect size *f* | A priori power | Total sample size |
|---|---|---|---|---|---|
| 2% variance | | | | | |
| All treatments | $F_{(4,25)}$ = 6246.6 | 0.99900 | 31.60696 | 99.9% | 10 (5 groups) |
| 15% variance | | | | | |
| All treatments | $F_{(4,25)}$ = 111.04 | 0.94672 | 4.21509 | 99.9% | 15 (5 groups) |
| 28% variance | | | | | |
| All treatments | $F_{(4,25)}$ = 31.868 | 0.83604 | 2.25807 | 99.9% | 20 (5 groups) |
| 40% variance | | | | | |
| All treatments | $F_{(4,25)}$ = 15.615 | 0.71416 | 1.58066 | 99.9% | 25 (5 groups) |

## Test family

- 2 tailed *t* test, difference between two independent means, Bonferroni's correction: alpha error = 0.0125

Power Calculations performed with G*Power software, version 3.1.7 (*Faul et al., 2007*).

## MAP3K8 values

| Group 1 | Group 2 | Effect size *d* | A priori power | Group 1 sample size | Group 2 sample size |
|---|---|---|---|---|---|
| 2% variance | | | | | |
| 1 µM PLX4720 | 1 µM PLX4720 + 1 µM CI1040 | 67.45484 | 99.9% | 2 | 2 |
| 1 µM PLX4720 | 1 µM PLX4720 + 1 µM AZD6244 | 55.51810 | 99.9% | 2 | 2 |
| 10 µM PLX4720 | 1 µM PLX4720 + 1 µM CI1040 | 46.68351 | 99.9% | 2 | 2 |
| 10 µM PLX4720 | 1 µM PLX4720 + 1 µM AZD6244 | 48.35265 | 99.9% | 2 | 2 |
| 15% variance | | | | | |

*Continued on next page*

*Continued*

| Group 1 | Group 2 | Effect size *d* | A priori power | Group 1 sample size | Group 2 sample size |
|---|---|---|---|---|---|
| 1 µM PLX4720 | 1 µM PLX4720 + 1 µM CI1040 | 7.24974 | 99.4% | 3 | 3 |
| 1 µM PLX4720 | 1 µM PLX4720 + 1 µM AZD6244 | 7.40241 | 99.6% | 3 | 3 |
| 10 µM PLX4720 | 1 µM PLX4720 + 1 µM CI1040 | 6.22447 | 97.2% | 3 | 3 |
| 10 µM PLX4720 | 1 µM PLX4720 + 1 µM AZD6244 | 6.44702 | 97.9% | 3 | 3 |
| 28% variance | | | | | |
| 1 µM PLX4720 | 1 µM PLX4720 + 1 µM CI1040 | 3.88379 | 93.0% | 4 | 4 |
| 1 µM PLX4720 | 1 µM PLX4720 + 1 µM AZD6244 | 3.96558 | 94.0% | 4 | 4 |
| 10 µM PLX4720 | 1 µM PLX4720 + 1 µM CI1040 | 3.33454 | 82.9% | 4 | 4 |
| 10 µM PLX4720 | 1 µM PLX4720 + 1 µM AZD6244 | 3.45376 | 85.7% | 4 | 4 |
| 40% variance | | | | | |
| 1 µM PLX4720 | 1 µM PLX4720 + 1 µM CI1040 | 2.71865 | 82.6% | 5 | 5 |
| 1 µM PLX4720 | 1 µM PLX4720 + 1 µM AZD6244 | 2.77591 | 84.3% | 5 | 5 |
| 10 µM PLX4720 | 1 µM PLX4720 + 1 µM CI1040 | 2.33418 | 81.5% | 6 | 6 |
| 10 µM PLX4720 | 1 µM PLX4720 + 1 µM AZD6244 | 2.41763 | 84.5% | 6 | 6 |

## Test family

- *F* test, ANOVA: Fixed effects, special, main effects and interactions, alpha error = 0.05.

Power Calculations performed with G*Power software, version 3.1.7 (*Faul et al., 2007*).
    ANOVA F test statistic and partial $\eta^2$ performed with R software, version 3.2.1 (*Team, 2015*).

## MEK1 and MEK1$^{DD}$ values

| Groups | F test statistic | Partial $\eta^2$ | Effect size *f* | A priori power | Total sample size |
|---|---|---|---|---|---|
| 2% variance | | | | | |
| All treatments | $F_{(4,50)}$ = 2716.2 | 0.99542 | 14.7409 | 99.9% | 20 (10 groups) |
| 15% variance | | | | | |
| All treatments | $F_{(4,50)}$ = 48.287 | 0.79437 | 1.96545 | 99.9% | 30 (10 groups) |
| 28% variance | | | | | |
| All treatments | $F_{(4,50)}$ = 13.858 | 0.52576 | 1.05292 | 99.9% | 40 (10 groups) |
| 40% variance | | | | | |
| All treatments | $F_{(4,50)}$ = 6.7904 | 0.35201 | 0.73704 | 98.7% | 50 (10 groups) |

In order to produce quantitative replication data, we will run the experiment four times. Each time we will quantify relative viability. We will determine the standard deviation of viability values across the biological replicates and combine this with the reported value from the original study to simulate the original effect size. We will use this simulated effect size to determine the number of replicates necessary to reach a power of at least 80%. We will then perform additional replicates, if required, to ensure that the experiment has more than 80% power to detect the original effect.

## Protocol 6

The original data presented is qualitative (images of Western blots). We used Image Studio Lite v. 4.0.21 (LI-COR) to perform densitometric analysis of the presented bands to quantify the original effect size where possible.

Summary of estimated original data reported in Figure 4F:

| Cell line | Drug(s) | Normalized pERK/ERK |
|---|---|---|
| MEK1 | DMSO | 1.0000 |
| | PLX4720 (1 µM) | 0.0799 |
| | PLX4720 (1 µM) + CI-1040 (1 µM) | 0.0021 |
| | PLX4720 (1 µM) + AZD6244 (1 µM) | 0.0002 |
| MEK1$^{DD}$ | DMSO | 1.0000 |
| | PLX4720 (1 µM) | 1.0203 |
| | PLX4720 (1 µM) + CI-1040 (1 µM) | 0.3015 |
| | PLX4720 (1 µM) + AZD6244 (1 µM) | 0.0166 |
| MAP3K8 | DMSO | 1.0000 |
| | PLX4720 (1 µM) | 0.7689 |
| | PLX4720 (1 µM) + CI-1040 (1 µM) | 0.0045 |
| | PLX4720 (1 µM) + AZD6244 (1 µM) | 0.0063 |

## Test family

- *F* test, ANOVA: Fixed effects, omnibus, one-way, alpha error = 0.05.

Power Calculations performed with G*Power software, version 3.1.7 (*Faul et al., 2007*).
ANOVA F test statistic and partial $\eta^2$ performed with R software, version 3.2.1 (*Team, 2015*).

## MAP3K8 values

| Groups | F test statistic | Partial $\eta^2$ | Effect size *f* | A priori power | Total sample size |
|---|---|---|---|---|---|
| 2% variance | | | | | |
| All treatments | F(3,8) = 5024.1 | 0.99947 | 43.4257 | 99.9% | 8 (4 groups) |
| 15% variance | | | | | |
| All treatments | F(3,8) = 89.317 | 0.97101 | 5.78735 | 99.9% | 8 (4 groups) |
| 28% variance | | | | | |
| All treatments | F(3,8) = 25.633 | 0.90577 | 3.10038 | 99.9% | 12 (4 groups) |
| 40% variance | | | | | |
| All treatments | F(3,8) = 12.560 | 0.82487 | 2.17026 | 99.9% | 16 (4 groups) |

## Test family

- 2 tailed *t* test, difference between two independent means, Bonferroni's correction: alpha error = 0.025

Power Calculations performed with G*Power software, version 3.1.7 (*Faul et al., 2007*).

## MAP3K8 values

| Group 1 | Group 2 | Effect size d | A priori power | Group 1 sample size | Group 2 sample size |
|---|---|---|---|---|---|
| 2% variance | | | | | |
| 1 µM PLX4720 | 1 µM PLX4720 + 1 µM CI1040 | 68.8057 | 99.9% | 2 | 2 |
| 1 µM PLX4720 | 1 µM PLX4720 + 1 µM AZD6244 | 70.5269 | 99.9% | 2 | 2 |
| 15% variance | | | | | |
| 1 µM PLX4720 | 1 µM PLX4720 + 1 µM CI1040 | 9.17409 | 87.8% | 2 | 2 |
| 1 µM PLX4720 | 1 µM PLX4720 + 1 µM AZD6244 | 9.40358 | 89.0% | 2 | 2 |
| 28% variance | | | | | |
| 1 µM PLX4720 | 1 µM PLX4720 + 1 µM CI1040 | 4.91469 | 95.5% | 3 | 3 |
| 1 µM PLX4720 | 1 µM PLX4720 + 1 µM AZD6244 | 5.03763 | 96.3% | 3 | 3 |
| 40% variance | | | | | |
| 1 µM PLX4720 | 1 µM PLX4720 + 1 µM CI1040 | 3.44028 | 93.7% | 4 | 4 |
| 1 µM PLX4720 | 1 µM PLX4720 + 1 µM AZD6244 | 3.52634 | 94.7% | 4 | 4 |

## Test family

- $F$ test, ANOVA: Fixed effects, special, main effects and interactions, alpha error = 0.05.

Power Calculations performed with G*Power software, version 3.1.7 (*Faul et al., 2007*).
ANOVA F test statistic and partial $\eta^2$ performed with R software, version 3.2.1 (*Team, 2015*).

## MEK1 and MEK1^DD values

| Groups | F test statistic | Partial $\eta^2$ | Effect size f | A priori power | Total sample size |
|---|---|---|---|---|---|
| 2% variance | | | | | |
| All treatments | $F(3,16) = 1847.2$ | 0.99712 | 18.6103 | 99.9% | 16 (8 groups) |
| 15% variance | | | | | |
| All treatments | $F(3,16) = 32.840$ | 0.86029 | 2.48143 | 99.9% | 16 (8 groups) |
| 28% variance | | | | | |
| All treatments | $F(3,16) = 9.4247$ | 0.63862 | 1.32934 | 99.9% | 24 (8 groups) |
| 40% variance | | | | | |
| All treatments | $F(3,16) = 4.6181$ | 0.46406 | 0.93053 | 99.0% | 32 (8 groups) |

In order to produce quantitative replication data, we will run the experiment three times. Each time we will quantify band intensity. We will determine the standard deviation of band intensity across the biological replicates and combine this with the reported value from the original study to simulate the original effect size. We will use this simulated effect size to determine the number of replicates necessary to reach a power of at least 80%. We will then perform additional replicates, if required, to ensure that the experiment has more than 80% power to detect the original effect.

## Acknowledgements

The Reproducibility Project: Cancer Biology core team would like to thank the original authors, in particular Cory Johannessen, for generously sharing critical information as well as reagents to ensure the fidelity and quality of this replication attempt. We thank Courtney Soderberg at the Center for

Open Science for assistance with statistical analyses. We would also like to thank the following companies for generously donating reagents to the Reproducibility Project: Cancer Biology; American Type Culture Collection (ATCC), Applied Biological Materials, BioLegend, Charles River Laboratories, Corning Incorporated, DDC Medical, EMD Millipore, Harlan Laboratories, LI-COR Biosciences, Mirus Bio, Novus Biologicals, Sigma-Aldrich, and System Biosciences (SBI).

## Additional information

### Group author details

Reproducibility Project: Cancer Biology

Elizabeth Iorns: Science Exchange, Palo Alto, United States; William Gunn: Mendeley, London, United Kingdom; Fraser Tan: Science Exchange, Palo Alto, United States; Joelle Lomax: Science Exchange, Palo Alto, United States; Nicole Perfito: Science Exchange, Palo Alto, United States; Timothy Errington: Center for Open Science, Palo Alto, United States

### Competing interests

VS, LY: Applied Biological Materials is a Science Exchange associated laboratory. RP:CB: EI, FT, JL, NP: Employed and hold shares in Science Exchange Inc. The other authors declare that no competing interests exist.

### Funding

| Funder | Author |
| --- | --- |
| Laura and John Arnold Foundation | Reproducibility Project: Cancer Biology |

The Reproducibility Project: Cancer Biology is funded by the Laura and John Arnold Foundation, provided to the Center for Open Science in collaboration with Science Exchange. The funder had no role in study design, data collection and interpretation, or the decision to submit the work for publication.

### Author contributions

VS, LY, MC, KO, Drafting or revising the article; RP:CB, Conception and design, Drafting or revising the article

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
