## [Decision Letter]

Thank you for submitting your work entitled "Registered report: COT drives resistance to RAF inhibition through MAP kinase pathway reactivation" for consideration by *eLife*. Your submission has been evaluated by Tony Hunter who has reviewed the protocols presented. We would ask to make to address the following comments before peer review:

1) In Protocol 4, the authors state "This experiment assesses the effect of the COT kinase inhibitor, TPL2, has on the MAPK pathway, in cells expressing elevated COT, as analyzed via Western blot." This leads me to believe that the authors are unaware that TPL2 is another name for COT, which is an ongoing confusion in the literature, and, from the start of the report, they need to clarify that COT and TPL2 are the same thing, both of which have the formal name MAP3K8. The authors of the paper being replicated obtained the TPL2/COT/MAP3K8 inhibitor from EMD Millipore, whereas the authors plan to obtain the TPL2/COT inhibitor from Santa Cruz rather than EMD Millipore. However, two compounds from the two companies have slightly different structures (i.e. EMD sells: 4-(cycloheptylamino)-6-(pyridin-3-yl-methylamino)-3-cyano-[1,7]-naphthyridine, whereas Santa Cruz sell: 4-(3-chloro-4-fluorophenylamino)-6-(pyridin-3-yl-methylamino)-3-cyano-[1,7]-naphthyridine, as their TPL2/COT inhibitor (sc-204351). This may not make a difference, but the authors will need to validate sc-204351 to show it has the expected ability to inhibit TPL2/COT kinase activity at the concentrations they are planning to use.

2) In Protocols 5 and 6, the authors plan to generate A375 cells expressing exogenous MEK1, MEK1^DD^ and MAP3K8/COT using lentivirus expression vector infection and blasticidin drug selection. However, they do not plan to ensure that the infected cells are actually expressing the kinase of interest at elevated levels, and this is obviously essential to do, so that the consequences of treatment with PLX4720 can be properly interpreted.

[Editors' note: further revisions were requested prior to acceptance, as described below.]

Thank you for submitting your work entitled "Registered report: COT drives resistance to RAF inhibition through MAP kinase pathway reactivation" for peer review at *eLife*. We apologize for the long delay in reaching this decision, but there was difficulty in finding qualified reviewers. Your submission has been favorably evaluated by Tony Hunter (Senior Editor) and two reviewers.

Reviewer 2 was satisfied, and therefore I have only included below the comments of Reviewer 1. As you will see the reviewer's comments mostly pertain to improving the quality of the experiments and the validity of the conclusions that can be drawn from them. Since the goal of a Registered Report is to replicate the original studies, you are not required to include the additional experiments, but if you believe that any of the suggested experiments would strengthen the conclusions and can readily be incorporated, perhaps as separate data panels, then you could do so. Please let me know your plans for proceeding with the experimental studies for the Registered Report.

Reviewer 1:

The Registered Report aims to repeat key experiments from the 2010 study of Johnanessen et al. which proposed that COT/TPL2 can drive the resistance of B-RAF[V600E] transformed cell lines to a selective RAF kinase inhibitor. COT/TPL2 was identified in an overexpression screen for genes that promote survival of A375 cultured with the specific RAF inhibitor PLX4720. This initial result is unsurprising given that is well established (in papers published over a decade earlier) that COT/TPL2 functions as a MKK1/2 kinase (similar to RAF kinases) and is constitutively active when overexpressed.

Proposed experiments:

The original paper proposed that overproduction of endogenous COT/TPL2 drives resistance of malignant melanomas to PLX4720. However, this conclusion was largely based on experiments with one malignant melanoma line, RPMI-7951 in which COT/TPL2 levels are relatively high. The impact of the study would have been greater if other RAF[V600E] transformed cell lines with COT/TPL2 overexpression had been analyzed in detail as per RPMI-7951 cells.

Protocol 1 (Figures 3B and 3E)

The effect of a range of concentrations of PLX4720 will be tested on the A375 and RPMI-7951 cell lines. Activation of the MKK1/2 > ERK1/2 MAP kinase pathway will be monitored by immunoblotting.

The original Figure 3E tests the effect of PLX4720 on A375, OUMS-23 and RPMI-7951. For the replication study, it is important that OUMS-23 cells, which express very high levels of COT/TPL2, are added to the proposed experiments to determine whether ERK1/2 phosphorylation is maintained in these cells after PLX4720 treatment. As noted above, a limitation of the original study was the dependence on very few cell lines, making generalization of results difficult.

Cell lysate will be resolved by SDS-PAGE and immunoblotted for pERK1/2, total ERK1/2, pMKK1/2, total MKK1/2, COT/TPL-2 and actin. From the protocol, three combinations of antibodies will be tested, presumably in three separate gels (although this is not clear). In our experience, probing blots with pERK1/2 antibody interferes with subsequent probing for total ERK1/2, even after antibody stripping. Consequently, it is better to probe for pERK1/2 and MKK1/2 on one blot and for pMKK1/2 and ERK1/2 on another.

It is planned to use HRP-coupled secondary antibodies and ECL for immunoblotting. However, the linear range of ECL is relatively small and it would be much better to quantify bands using IR Dye-labeled secondary antibodies and a LI-COR scanner. This was also true when the original experiments were carried out.

Protocol 3

This protocol will test the effect of PLX4720 on the viability of A375 and RPMI-7951 cells. However, the experiment shown in Figure 3D of the original paper tested the effect of PLX4720 on a panel of cells lines, with low or high COT/TPL2 expression. Since the authors are making general claims about the role of COT/TPL2 in resistance to RAF inhibitors, OUMS-23 should be analyzed as well. The inclusion of a cell line with low COT/TPL2 levels, similar to A375, would also be preferable.

Protocol 4

The authors have used a Wyeth COT/TPL2 small molecule inhibitor to determine whether COT/TPL2 signaling is required for MKK1/2 and ERK1/2. Our experience some of the early Wyeth compounds is that they have clear off-target effects (revealed by testing their effect on cytokine production in LPS-stimulated *Map3k8*-/- macrophages). The significance of the experiment shown in Figure 3I therefore is highly questionable. In addition, since the effects of the inhibitor on are only partial, pMKK1/2 and pERK1/2 levels in the original study should have been quantified from multiple experiments and differences tested for statistical significance (as noted in the Registered Report proposal). In view of the limitations of the COT/TPL2 inhibitor experiment, it is important that the siRNA knockdown experiment in Figure 3H is repeated. For this, it is essential to correlate reduction in pERK1/2 levels with the degree of COT/TPL2 knockdown achieved with shCOT-1 and shCOT-2.

See comments on Protocol 2 regarding lysis buffer, immunoblotting etc.

Comment

In Figure 4C, why are A375 cells expressing MEK1^DD^ sensitive to RAF265 inhibitor? Similarly, with SKMEL28 cells expressing MEK1^DD^. I could find no reference to these results in the text. I would have expected MEK1^DD^ expression in A375 cells to bypass the requirement for constitutive RAF signaling. Am I missing something?

---

## [Author Response]

1) In Protocol 4, the authors state "This experiment assesses the effect of the COT kinase inhibitor, TPL2, has on the MAPK pathway, in cells expressing elevated COT, as analyzed via Western blot." This leads me to believe that the authors are unaware that TPL2 is another name for COT, which is an ongoing confusion in the literature, and, from the start of the report, they need to clarify that COT and TPL2 are the same thing, both of which have the formal name MAP3K8. The authors of the paper being replicated obtained the TPL2/COT/MAP3K8 inhibitor from EMD Millipore, whereas the authors plan to obtain the TPL2/COT inhibitor from Santa Cruz rather than EMD Millipore. However, two compounds from the two companies have slightly different structures (i.e. EMD sells: 4-(cycloheptylamino)-6-(pyridin-3-yl-methylamino)-3-cyano-[1,7]-naphthyridine, whereas Santa Cruz sell: 4-(3-chloro-4-fluorophenylamino)-6-(pyridin-3-yl-methylamino)-3-cyano-[1,7]-naphthyridine, as their TPL2/COT inhibitor (sc-204351). This may not make a difference, but the authors will need to validate sc-204351 to show it has the expected ability to inhibit TPL2/COT kinase activity at the concentrations they are planning to use.

Thank you for the suggestion about the protein name. We have gone through the manuscript and defined in the Introduction all the names used for the protein of interest and for consistency are using the formal name MAP3K8 from that point on. We used the same name that was used in the original publication for each experiment/figure/reagent in an attempt to make the comparison easier to follow, but agree it is better to not add to the ongoing confusion in the literature.

Regarding the inhibitor, we have switched the inhibitor that will be used to the EMD source to minimize differences from the original experiment. However, the EMD source originally used, and now planned to be used in the replication attempt, is the same compound as the Santa Cruz source we had originally suggested (4-(3-chloro-4-fluorophenylamino)-6-(pyridin-3-yl-methylamino)-3-cyano-[1,7]-naphthyridine; CAS number: 871307-18-5. EMD also has the type II MAP3K8 kinase inhibitor (4-(cycloheptylamino)-6-(pyridin-3-yl-methylamino)-3-cyano-[1,7]-naphthyridine; CAS number: 1186649-59-4), however this molecule is not being utilized in the replication attempt as it was not used in the original experiment.

*2) In Protocols 5 and 6, the authors plan to generate A375 cells expressing exogenous MEK1, MEK1^DD^ and MAP3K8/COT using lentivirus expression vector infection and blasticidin drug selection. However, they do not plan to ensure that the infected cells are actually expressing the kinase of interest at elevated levels, and this is obviously essential to do, so that the consequences of treatment with PLX4720 can be properly interpreted.*

The expression of the exogenous proteins will be assessed by western blot using an antibody against the V5 tag, which is included in the vectors for expression of MEK1, MEK1^DD^, and MAP3K8. We have more clearly described this in the revised manuscript. Additionally, in the revised manuscript we are also including antibodies against MAP3K8 and MEK1 to assess the relative degree of elevated protein levels by western blot.

[Editors' note: further revisions were requested prior to acceptance, as described below.]

Reviewer 2 was satisfied, and therefore I have only included below the comments of Reviewer 1. As you will see the reviewer's comments mostly pertain to improving the quality of the experiments and the validity of the conclusions that can be drawn from them. Since the goal of a Registered Report is to replicate the original studies, you are not required to include the additional experiments, but if you believe that any of the suggested experiments would strengthen the conclusions and can readily be incorporated, perhaps as separate data panels, then you could do so. Please let me know your plans for proceeding with the experimental studies for the Registered Report. Reviewer 1:

Proposed experiments: The original paper proposed that overproduction of endogenous COT/TPL2 drives resistance of malignant melanomas to PLX4720. However, this conclusion was largely based on experiments with one malignant melanoma line, RPMI-7951 in which COT/TPL2 levels are relatively high. The impact of the study would have been greater if other RAF[V600E] transformed cell lines with COT/TPL2 overexpression had been analyzed in detail as per RPMI-7951 cells. Protocol 1 (Figures 3B and 3E) The effect of a range of concentrations of PLX4720 will be tested on the A375 and RPMI-7951 cell lines. Activation of the MKK1/2 > ERK1/2 MAP kinase pathway will be monitored by immunoblotting.

*The original Figure 3E tests the effect of PLX4720 on A375, OUMS-23 and RPMI-7951. For the replication study, it is important that OUMS-23 cells, which express very high levels of COT/TPL2, are added to the proposed experiments to determine whether ERK1/2 phosphorylation is maintained in these cells after PLX4720 treatment. As noted above, a limitation of the original study was the dependence on very few cell lines, making generalization of results difficult.*

Thank you for this suggestion. We agree and have added the OUMS-23 cell line in the revised manuscript. We have also adjusted the analysis plans and power calculations to reflect the analysis of OUMS-23 cells similar to A375 and RPMI-7951.

*Cell lysate will be resolved by SDS-PAGE and immunoblotted for pERK1/2, total ERK1/2, pMKK1/2, total MKK1/2, COT/TPL-2 and actin. From the protocol, three combinations of antibodies will be tested, presumably in three separate gels (although this is not clear). In our experience, probing blots with pERK1/2 antibody interferes with subsequent probing for total ERK1/2, even after antibody stripping. Consequently, it is better to probe for pERK1/2 and MKK1/2 on one blot and for pMKK1/2 and ERK1/2 on another.*

We have clarified the protocols where multiple gels are run to indicate the intention of running three separate gels. We have also changed the probing of the blots to reflect this strategy. As a result, we have also changed the cat # of the total ERK1/2 and MEK1/2 antibodies to a different species (opposite what pERK1/2 and pMEK1/2 are) in addition to the normalization strategy to reflect what is probed on the same blot.

*It is planned to use HRP-coupled secondary antibodies and ECL for immunoblotting. However, the linear range of ECL is relatively small and it would be much better to quantify bands using IR Dye-labeled secondary antibodies and a LI-COR scanner. This was also true when the original experiments were carried out.*

We agree that IR Dye-labeled secondary antibodies and a LI-COR scanner are ideal and the linear range of ECL is relatively small. While the lab is not equipped with a LI-COR scanner we will make all the exposures from ECL publicly available. There is also the possibility the lab might be able to access a nearby scanner. We have added a statement in the manuscript to indicate that if a LI-COR scanner is available it will be used instead of the ECL approach, with the HRP coupled secondary antibodies switched for the appropriate IR Dye-labeled secondary antibodies. This does not impact the proposed analysis plan. If the antibody detection is performed with the LI-COR scanner, it will also be made transparent in the Replication Study.

*Protocol 3 This protocol will test the effect of PLX4720 on the viability of A375 and RPMI-7951 cells. However, the experiment shown in Figure 3D of the original paper tested the effect of PLX4720 on a panel of cells lines, with low or high COT/TPL2 expression. Since the authors are making general claims about the role of COT/TPL2 in resistance to RAF inhibitors, OUMS-23 should be analyzed as well. The inclusion of a cell line with low COT/TPL2 levels, similar to A375, would also be preferable.*

Thank you for this suggestion. We agree and have added the OUMS-23 cell line in the revised manuscript for Protocol 2 as well as Protocol 3. We have also adjusted the analysis plans and power calculations to reflect the analysis of OUMS-23 cells similar to A375 and RPMI-7951.

Protocol 4 The authors have used a Wyeth COT/TPL2 small molecule inhibitor to determine whether COT/TPL2 signaling is required for MKK1/2 and ERK1/2. Our experience some of the early Wyeth compounds is that they have clear off-target effects (revealed by testing their effect on cytokine production in LPS-stimulated Map3k8-/- macrophages). The significance of the experiment shown in Figure 3I therefore is highly questionable. In addition, since the effects of the inhibitor on are only partial, pMKK1/2 and pERK1/2 levels in the original study should have been quantified from multiple experiments and differences tested for statistical significance (as noted in the Registered Report proposal). In view of the limitations of the COT/TPL2 inhibitor experiment, it is important that the siRNA knockdown experiment in Figure 3H is repeated. For this, it is essential to correlate reduction in pERK1/2 levels with the degree of COT/TPL2 knockdown achieved with shCOT-1 and shCOT-2.

*See comments on Protocol 2 regarding lysis buffer, immunoblotting etc.*

We agree that including the shCOT experiment would be of interest considering the limitations of the COT/TPL2 inhibitor experiment and by excluding it limits the scope of what can be interpreted from this replication attempt, but we are attempting to identify a balance of breadth of sampling for general inference with sensible investment of resources on replication projects. While the COT/TPL2 small molecular inhibitor has off-target effects, our anticipation is that the proposed quantification and statistical tests will further evaluate whether this inhibitor has an effect on pMEK1/2 and pERK1/2 levels, whether directly or indirectly through COT/TPL2.

*Comment In Figure 4C, why are A375 cells expressing MEK1^DD^ sensitive to RAF265 inhibitor? Similarly, with SKMEL28 cells expressing MEK1^DD^. I could find no reference to these results in the text. I would have expected MEK1^DD^ expression in A375 cells to bypass the requirement for constitutive RAF signaling. Am I missing something?*

We agree with the reviewer that this seems to be an interesting observation made by the original lab even though it wasn’t discussed and should be investigated by future replication efforts.